# Enzymatic biofilm digestion in soil aggregates facilitates the release of particulate organic matter by sonication

**F. Büks[1] and M. Kaupenjohann[1]**

[1] Chair of Soil Science, Department of Ecology, Technische Universität Berlin.

*Correspondence to:* F. Büks (frederick.bueks@tu-berlin.de)

**Abstract.**

The stability of soil aggregates against shearing and compressive forces as well as water caused dispersion is an integral marker of soil quality. High stability results in less compaction and erosion and has been linked to enhanced water retention, dynamic water transport and aeration regimes, increased rooting depth and protection of soil organic matter (SOM) against microbial degradation. In turn, particulate organic matter is supposed to support soil aggregate stabilization. For decades the importance of biofilm extracellular polymeric substances (EPS) regarding particulate organic matter (POM) occlusion and aggregate stability has been canonical because of its distribution, geometric structure and ability to link primary particles. However, experimental proof is still missing. This lack is mainly due to methodological reasons. Thus, the objective of this work is to develop a method of enzymatic biofilm detachment for studying the effects of EPS on POM occlusion. The method combines an enzymatic pre-treatment with different activities of α-glucosidase, β-galactosidase, DNAse and lipase with a subsequent sequential ultrasonic treatment for disaggregation and density-fractioning of soils. Particulate organic matter releases of treated samples were compared to an enzyme-free control. To test the efficacy of biofilm detachment the ratio of bacterial DNA from suspended cells and the remaining biofilm after enzymatic treatment were measured by quantitative real-time PCR. Although the enzyme treatment was not sufficient for total biofilm removal, our results indicate that EPS may attach particulate organic matter (POM) within soil aggregates. The tendency to additional POM release with increased application of enzymes was attributed to a slight loss in aggregate stability. This suggests that an effect of agricultural practices on soil microbial populations could influence POM occlusion/aggregate stability and thereby carbon cycle/soil quality.

# 1 Introduction

Soil organic matter (SOM) comprises 50% (~1,700 Gt, including peat) of the near-surface terrestrial carbon budget, compared to ~813 Gt bound in the atmosphere (Lal, 2008). Beside carbon storage and its influence on the atmospheric $CO_2$ balance, manifold ecological soil functions are mediated by different SOM types like dissolved organic matter (DOM), particulate organic matter (POM), molecular organic matter of organo-mineral associations, colloidal organic matter and coprecipitated molecular organic matter (Kalbitz et al., 2000; Weng et al., 2002; Pokrovsky et al., 2005; Eusterhues et al., 2008). For example, POM is a structural component of soil aggregates, a nutrient source and provides surfaces for microbial growth  (Chenu and Stotzky, 2002; Bronick and Lal, 2005). Parts of the POM are occluded within soil aggregates (Six et al., 2002). Physical isolation protects POM against microbial degradation (Six et al., 2002; Lützow et al., 2006) and maintains its ecological functions, while on the other hand this POM is thought to promote soil aggregation (Bronick and Lal, 2005). Therefore, many benefits of soil POM are linked to soil aggregate stability.

The stability of soil aggregates against shear and compression forces *(Skidmore and Powers, 1982)* as well as disaggregation caused by water (Tisdall and Oades, 1982) is an integral marker of soil quality (Bronick and Lal, 2005). Since aggregate stability implies pore stability, it results in less soil compactibility (Baumgartl and Horn, 1991; Alaoui et al., 2011) and a more dynamic water transport regime in the macropores that reduces erosion caused by surface runoff (Barthes and Roose, 2002). Other benefits in comparison to compacted soils are a higher aeration (Ball and Robertson, 1994) and lower penetration resistance (Bennie and Burger, 1988) causing increased rootability and rooting depth (Bengough and Mullins, 1990; Taylor and Brar, 1991). In addition, micropores within the aggregates enhance water retention.

The occlusion of POM within soil aggregates depends on the properties of the aggregated components. The mineral part of the solid soil matrix is composed of siliceous sand, silt and clay particles, oxides and hydroxides of Fe, Al and Mn as well as diverse minor minerals. Sticking together, pervaded and coated with multivalent cations and organic constituents like soluble metabolic products, humic substances, black carbon and other POM, macro-aggregates (>250 µm) are formed by direct coagulation or built of micro-aggregates (<250 µm). (Bronick and Lal, 2005; Brodowski et al., 2006; Lützow et al., 2006)







The structure-bearing primary particles, precipitates and adsorbed molecules cohere by physico-chemical interactions between (i) permanent charge of mainly the clay mineral fraction, (ii) multivalent cations with small hydrate shells such as $Ca^{2+}$, $Fe^{3+}$ and $Al^{3+}$, (iii) variable charges of various minerals and SOM and (IV) variable and permanent dipoles of different soil components. Also carbonates, phosphates and other microbial precipitates force up aggregation and occlusion of POM. (Jastrow and Miller, 1997; Bronick and Lal, 2005)

In addition, since a few decades biological structures like bacterial colonies, bacterial pseudomycelia, algae, fungal hyphae and their exudates (e.g. glomalin), roots and soil fauna are accepted as a major factor of soil aggregation (Tisdall, 1991; Oades, 1993; Wright and Upadhyaya, 1998; Brown et al., 2000; Chenu and Stotzky, 2002; Rillig, 2004; Bronick and Lal, 2005). Furthermore the role of extracellular polymeric substance (EPS) of bacterial biofilms as an adhesive between soil particles is seen to be of importance (Baldock, 2002; Ashman et al., 2009).

Physical and chemical properties of soil mineral and organic matter allow to hypothesize a simple spacial model of the inner geometry of soil aggregates, that includes biofilms as links between primary particles (Fig. 1).

The biofilm itself is a viscous microenvironment mainly built up of 90-97% water (Zhang et al., 1998; Schmitt and Flemming, 1999; Pal and Paul, 2008). The remaining dry mass contains differing ratios of polysaccharides, extracellular DNA (eDNA), proteins and lipids besides 10-50% cell biomass (More et al., 2014). In contrast to 'biofilm', EPS terms the extracellular polymeric matrix excluding cells. Extracellular polysaccharides cause the EPS structural stability by means of entanglement and $Ca^{2+}$ bridging between molecules. So does eDNA (Das et al., 2014). Proteins function as enzymes and structural links stabilizing the polysaccharide matrix, while lipids act as biosurfactants for bacterial attachment on surfaces. (Flemming and Wingender, 2010)

The composition of EPS is highly variable depending on community composition and environmental cues (Table 1): Redmile-Gordon et al. (2014) measured a natural habitat extracellular polysaccharide concentration of 401 µg $g^{-1}$ dry soil in grassland and 169 µg $g^{-1}$ in fallows. Diverse single- and multi-species biofilms show a proportion of polysaccharides on dry EPS of up to 95% (Pal and Paul, 2008; More et al., 2014). Different single- and multi-species biofilms in laboratory cultures and natural soils have a dry EPS eDNA content up to 10% (More et al., 2014). For forest soils values of 1.95 up to 41.1 µg $g^{-1}$ dry soil are known

(Niemeyer and Gessler, 2002; Agnelli et al., 2004). Extracellular DNA concentration of other diverse soils ranges between 0.03 and 200 µg g$^{-1}$ dry soil (Niemeyer and Gessler, 2002; Pietramellara et al., 2009), whereas concentrations in soils explicitly used for agriculture are unknown. Extracellular matrix protein concentration was measured at 163 µg g$^{-1}$ dry soil in grassland and 43 µg g$^{-1}$ dry soil in fallow (Redmile-Gordon et al., 2014), but can contribute the largest fraction of EPS dry mass, e.g. 60% (More et al., 2014), and even up to 75% in *P. putida* biofilms in laboratory cultures (Jahn et al., 1999). The typical proportion of lipids in the EPS dry-mass of different non-soil biofilms amounts up to 10% (More et al., 2014). Sparse molar mass data from different environments comprise $0.5 \times 10^6$ to $2 \times 10^6$ Da for polysaccharides (Flemming and Wingender, 2010), $7.75 \times 10^4$ to $2.32 \times 10^7$ Da for eDNA (DeFlaun et al., 1987) and 750 to 1,500 Da for lipids (Munk, 2008).

The extracellular matrix is not only exuded by soil bacteria and archaea, but also by fungi and algae. It is engineered by grazing protozoa and small metazoa as well as microbial extracellular enzymes. (Battin et al., 2007; Flemming and Wingender, 2010)

The activity of EPS degrading enzymes in natural soils spans up to two orders of magnitude: The α-glucosidase and β-galactosidase acitivity of various soils ranges from 0.00011 U g$^{-1}$ to 0.0011 U g$^{-1}$ and from 0.00017 to 0.0094 U g$^{-1}$, respectively (Eivazi and Tabatabai, 1988; Acosta-Martinez and Tabatabai, 2000). The lipase activity in coarse mineral soils shows values from 0.3 U g$^{-1}$ in a sandy soil (Cooper and Morgan, 1981) to 2.09 U g$^{-1}$ in a Luvisol (Margesin et al., 2000) and up to 5 U g$^{-1}$ in a Leptosol (Margesin et al., 1999). Data for eDNAse activity in soils are not available.

Not much is known about the contribution of EPS to POM occlusion and aggregate stability in relation to other aggregate stabilizing factors. That is mainly due to methodological reasons: Though e.g. Tang et al. (2011) showed a significant contribution of bacterial growth on aggregate stability, the observations could not definitely be attributed to soil microbial exopolysaccharide production. Redmile-Gordon et al. (2014) subsequently found that the techniques previously used to measure extracellular polysaccharides in soil co-extracted large quantities of 'random' soil organic matter which confounded estimates of EPS production. Owing to the widespread interest in the role of biofilms on soil fertility, the objectives of this work are (i) to design a selective method for enzymatic biofilm detachment with minor impact on other types of aggregate bonds and (ii) to apply the method to an agricultural soil to provide indications of the influence of biofilm cohesion on POM fixation, which is expected to contribute to aggregate stability (Six et al.,

2004).

The method combines a modified enzymatic pre-treatment (Böckelmann et al., 2003) with α-glucosidase, β-galactosidase, DNAse and lipase, a determination of the DNA ratio of sessile to suspended cells after enzymatic treatment and an ultrasonication of soil aggregates followed by density-fractioning and soil organic carbon (SOC) measurement (Kaiser and Berhe, 2014). The ultrasonication/density-fractionation separates SOC into
three operational solid fractions: non-occluded free light fraction SOC (fLF-SOC), aggregate-embedded occluded light fraction SOC (oLF-SOC) and colloidal as well as (macro)molecular SOC, which is not detachable from mineral surfaces by the chosen fractioning method and subsumed under heavy fraction (HF-SOC) (Kaiser and Berhe, 2014).

We hypothesize that a destabilization of the EPS matrix occurs during enzymatic treatment. This should result in an increased cell detachment from aggregates. We also expect an increased fLF-SOC release from destabilized aggregates compared to the control and a shift of the oLF-SOC ratio from higher to lower binding strength (represented by ultrasonic energy levels) that is interpretable as alteration of soil aggregate stability.

## 2 Materials and methods

### 2.1 Soil properties and microbial biomass

Well aggregated silty sand (Su3) of a plowed topsoil from a cropland near Berge (Brandenburg/Germany) was air-dried and sieved to obtain a particle size of 0.63 to 2.0 mm containing mainly macro-aggregates. The aggregates have a $pH_{CaCl2}$ of 6.9, $C_{org}$ of 8.7 mg $g^{-1}$ and a carbonate concentration of 0.2 mg $g^{-1}$.

To estimate the soil microbial biomass, first 8 x 10 g of soil aggregates have been adjusted to 70 vol% soil water content and incubated for 70 hours at 20°C in the dark to attain basal respiration. Then, based on DIN EN ISO 14240-2 half of the samples were fumigated with ethanol-free chloroform in an evacuated desiccator for 24 h, whereas the other half remained untreated. Afterwards chloroform was removed and both halves were extracted with 40 ml of 0.5 M $K_2SO_4$ solution by 30 min of horizontal shaking and filtered through 0.7 µm glass fiber filters. The DOC concentrations of all filtrates were measured by a TOC Analyzer (TOC-5050A, Shimadzu). 176 ± 22 µg microbial carbon $g^{-1}$ dry soil ($C_{mic}$) were derived from the difference between DOC concentrations of fumigated and non-fumigated samples multiplied by a conversion factor of 2.22 (Joergensen, 1996). Soil bacterial biomass was derived from $C_{mic}$ as 352 ± 44 mg $kg^{-1}$ assuming 0.5 as a ratio of $C_{mic}$ to total cell dry mass (Bratbak and Dundas, 1984).

### 2.2 Detachment scenarios

Four degradative enzymes were selected on the basis of soil pH and temperature used for catalytic unit definition ($T_{def}$): α-glucosidase from *S. cerevisiae* (Sigma-Aldrich, $pH_{opt}$ 6 to 6.5, $T_{def}$=37°C, product number G0660) hydrolyzes terminal α-1,4-glycosidic linkages in polysaccharides as β-galactosidase from *E. coli* (Sigma-Aldrich, $pH_{opt}$ 6 to 8, $T_{def}$=37°C, product number G5635) does with β-glycosidic bonds. Lipase from porcine pancreas (Sigma-Aldrich, $pH_{def}$ 7.7, $T_{def}$=37°C, product number L0382) splits fatty acids from lipids via hydrolysis, but do not digest phospholipids, which are part of bacterial membranes. DNAse I from bovine pancreas ($pH_{def}$ 5, $T_{def}$=25°C, product number D5025) breaks the phosphodiester linkages between nucleotides of DNA as an endonuclease. Proteases were not used because of their promiscuity and therefore incalculable influence on the other applied enzymes.

Literature shows a wide range of target concentrations related to these enzymes in different soils. As we do not know target concentrations of our soil (due to a lack of extraction methods), we considered the largest published values (Table 2) of EPS content $\left(\xi_{EPS}^{max}\right)$ and enzyme target dry mass contents $\left(\xi_{target}^{max}\right)$ from literature. Further, as bacterial dry mass $\left(\xi_{cell}^{min}\right)$ and target molar masses $\left(M_{target}^{min}\right)$ vary as well, here we choose the minimum percentage and the smallest mass, respectively. These values conduce to a "worst-case" point of view with a maximum of enzyme targets. Any other boundary conditions such as ion activity, diffusion rates or metabolization of enzymes by soil organisms were disregarded.

Calculated by Eq. (1)

$$Unit_{target} = \frac{c_{cell} \cdot q \cdot \xi_{EPS}^{max} \cdot \xi_{target}^{max} \cdot m_{sample}}{\xi_{cell}^{min} \cdot M_{target}^{min} \cdot t} \quad (1)$$

with variables listed in Table 2 and Table 3, sufficient enzymes were provided to digest the expected EPS concentration in five scenarios: In the E1 scenario $c_{cell}$ was given by the results of fumigation-extraction. In the E2 scenario a bacterial dry mass of 500 g m$^{-2}$ in the upper 30 cm is considered, which is assumed to be the maximum for middle and northern European soils (Brauns, 1968). Supposing a soil bulk density of 1.4 g cm$^{-3}$, a $c_{cell}$ of 1190.5 µg g$^{-1}$ dry soil is given. Although the soil bulk density of the soil aggregate samples is ~1.15 g/cm$^3$, we decided to use the soil bulk density of the original soil, which is in the normal range of sandy silk soil (~1.40 g/cm³) (Chaudhari et al., 2013). This is due to the fact that biofilm populations are mentioned to be mainly located in soil aggregates (Nunan et al., 2003) and accords to the "worst-case"-approach. The E3 scenario uses a 100-fold excess (q=100, Table 3) of the enzyme activities applied in the E2 scenario, whereas the E4 scenario contained the 2,820-fold, which is slightly higher than activities used in Böckelmann et al. (2003). Enzyme-free samples (E0) were used as a control.

## 2.3 Release of POM carbon

Fifteen g of air-dried soil aggregates were incubated in 5 replicates per scenario with 3.4 ml of highly concentrated artificial rainwater (ARW: 0.2 mM NH$_4$NO$_3$, 0.3 mM MgSO$_4$ x 7H$_2$O, 0.5 mM CaCl$_2$ x 2H$_2$O, 0.5 mM Na$_2$SO$_4$, 15 mM KCl, pH 5.7) for 3 days at 20°C in

the dark to establish basal respiration and avoid slaking in the following preparation steps. After incubation 2.5 ml of ARW containing enzymatic units according to Table 3 were added to the samples. By means of a following incubation at 37°C, enzymes were let to work near their catalytic optimum for 1h, which is proven to be sufficient for biofilm degradation (Böckelmann et al., 2003). After this enzymatic pretreatment, 67.2 ml of 1.67 g cm$^{-3}$ dense sodium polytungstate (SPT) solution were added resulting in a density cut-off of 1.6 g/cm³, and samples were stored for 30 minutes to allow SPT diffusion into the aggregates. Then samples were centrifuged for 26 min with 3,569 G. Sodium polytungstate solution with floating fLF was filtered through an 1.5 µm pore size glass fibre filter to capture LF particles. Afterwards following Golchin et al. (1994) aggregate samples were consecutively disaggregated in four steps by application of each 50 J ml$^{-1}$ of ultrasonic energy (Branson© Sonifier 250) for 1 min 15 sec. The energy output was determined by measuring the heating rate of water inside a dewar vessel (Schmidt et al., 1999). Every treatment cycle consisted of ultrasonication, centrifugation for 26 min with 3,569 G and filtering of SPT solution through an 1.5 µm pore size glass fibre filter to capture the LF. Afterwards the LFs and the remaining soil matrix ('sediment', consisting of oLF bonded >150 J ml$^{-1}$ and the HF) were frozen, lyophilized, ground and dried at 105°C. Total amount of fraction carbon was determined using an Elementar Vario EL III CNS Analyzer and the absence of carbonates was proved, respectively.





## 2.4 Release of bacterial DNA

The release of bacterial cells into the solution was quantified using a FastDNA™ SPIN Kit for Soil and quantitative real-time PCR.


Therefor 45 µl of ARW were added directly to 0.1 g of air-dried aggregates. The samples were sterilely incubated in duplicate at 20°C for 3 days in the dark in a closed FastPrep Lysing Matrix E tube during run to basal respiration. Then 30 µl of ARW containing enzymatic units according to Table 3 were distributed equally to the aggregates' surfaces. The samples were incubated for 1 h at 37°C in a heating block, cooled down on ice to decrease enzyme activity and washed three times in 1 ml of ARW not by shaking but gently rotating along the tube's longitudinal axis to separate detached and planktonic cells from the soil matrix. Supernatants of all three washing steps were removed carefully with a pipette, pooled and centrifuged at 13.000 G for 15 minutes at 4°C. Then the supernatant


was discarded, the pallet resuspended in 200 µl ARW and transfered to another FastPrep Lysing Matrix E tube. Both soil and washing ARW samples were extracted and purified at 4°C following the FastDNA™ SPIN Kit for Soil manual. All DNA samples were stored at -20°C for further use. A direct subsampling from the aggregate stability experiment was rejected due to its destructive capability regarding aggregates. Temperature, substrate, pH

and water content of the DNA experiment were similar to the incubation of samples for the measurement of aggregate stability. Further differences (e.g. soil volume) were disregarded.

Amplification of 10-fold diluted DNA samples was performed using a C1000 Touch Thermal Cycler (BioRad). According to the reference for SG qPCR Master Mix (Roboklon)

thermocycling comprised an initial denaturation at 95°C for 10 min as well as 55 cycles of 15 sec of denaturation at 95°C, 20 sec of annealing at 49°C and 30 sec of elongation at 72°C. The reaction mix contained 1 µl PCR-$H_2O$, 12.5 µl SG qPCR MasterMix, each 0.75 µl of a 20 µmol $l^{-1}$ solution of the universal bacterial primers 63f (5'-CAGGCCTAACACATGCAAGTC-3') and 341r (5'-CTGCTGCCTCCCGTAGG-3') (Muyzer

et al., 1993; Marchesi et al., 1998) and 10 µl template DNA. *Escherichia coli* 16s DNA solution containing 10,000 copies $\mu l^{-1}$ was used as qPCR standard in steps of tenfold diluted concentration from $10^6$ to $10^2$ copies $\mu l^{-1}$.

**2.5 Statistics**

For evaluation of the light fraction SOC (LF-SOC) release, mean values as well as standard deviations were calculated. Parallels of each variant were positively tested to provide normal distribution and evidence of variance homogeneity (Shapiro Wilk test, Levene test, both p>0.05, data not shown). One way analysis of variance (ANOVA) was applied followed by Tukey test to clarify significant (p<0.05) differences in LF-SOC release

between variants of each energy level. Results of bacterial DNA release were presented as duplicates.


# 3 Results

## 3.1 Release of POM carbon

The relative LF carbon release from soil aggregate samples after different enzymatic treatments is shown in Fig. 2. The proportionate C of each captured fraction is defined as $C_{frac}\ C_\Sigma^{-1}$, in which $C_{frac}$ is the release of LF-SOC per energy level or – in case of the sediment – the organic carbon remaining in the soil matrix. $C_\Sigma$ is the total SOC of all separated LFs and the sediment.

Averaging all treatments, around 79% of $C_\Sigma$ remain in the sediment, whereas the bulk of LF-SOC is released as weakly bound oLF (50 J ml$^{-1}$) and fLF. Only around 4.5% of $C_\Sigma$ is detached at 100 J ml$^{-1}$ and 150 J ml$^{-1}$.

None of the enzymatic treatments altered the quantity of fLF-SOC released in the absence of sonication (0 J ml$^{-1}$).

In contrast, visible differences to the control were shown for E1 (decrease, p=0.34) and E4 (increase, p=0.42) at mild sonication (50 J ml$^{-1}$), whereas E2 (p=1.00) and E3 (p=1.00) are very similar to the control. The difference between E1 and E4 was statistically significant (p=0.01) as indicated by the Tukey test, and the addition of the highest enzyme concentration (E4) caused the release of about 63% more oLF-SOC than occurred with the addition of the lowest concentration (E1).

Released LF-SOC at 100 and 150 J ml$^{-1}$ is not different among treatments. Only the E2 scenario shows any tendency of increased oLF-SOC release at 100 J ml$^{-1}$ compared to the other treatments (p=0.07 compared to E3).

The sediment represents the SOC remaining unextractable at ≤150 J ml$^{-1}$ and accordingly shows a trend to decrease with increasing enzyme activity. In relation to the control, nearly the whole alteration in the oLF-SOC releases of E1 and E4 at 50 J/ml as well as E2 at 100 J/ml comes from the sediment fraction, but hardly from the other LFs. However, opposite reallocation of SOC between fractions due to converse physico-chemical effects can only be observed in sum. Therefore alterations must be considered as net C transfer between stability fractions.

Cumulating LF-SOC releases of all energy levels, E1 shows a reduction by 16% compared to the control (3.3% of $C_\Sigma$), whereas E4 was increased by 10% (2.2% of $C_\Sigma$). The strongest enzymatic treatment (E4) caused the release of about 58% (0.49 mg/g dry soil) more

cumulated LF-SOC than occurred with scenario E1.

## 3.2 Release of bacterial DNA

The relative DNA release after enzymatic treatment, as pictured with the treatments E0, E1 and E4 in Fig. 3, is defined as the ratio of extracted DNA from suspended bacterial cells

($DNA_{susp}$) to the sum of DNA extracted from suspended and sessile bacterial cells and the remaining EPS ($DNA_{\Sigma}$) multiplied by 100. While there was no difference in relative DNA release in the wash of control and low enzyme additions, treatment E4 caused an increase to more than double the DNA content of either E0 or E1, which amounts to 5.6% of total DNA. This increase is caused by both an increase in released bacterial DNA from

suspended bacterial cells and a decrease in eDNA remaining in washed soil.




## 4 Discussion

We found that increasing the quantity of enzymes applied to aggregates led to increased release of LF-SOC when aggregates were sonicated. This detachment is explained by the following mechanism: The enzyme mix flows into the unsaturated pore space. From there α-glucosidase, β-galactosidase, DNAse and lipase diffuse into the biofilm matrix, where structural components like polysaccharides, eDNA and lipids are digested as approved for diverse enzymes and enzyme targets in ecological and medical studies (Böckelmann et al., 2003; Walker et al., 2007). We propose a simple spacial model to explain the observed findings: The biofilm bridges gaps between organic and mineral primary particles, connects them in addition to other physico-chemical bondings and builds a restructured pore system inside the aggregate (Fig. 1). As macromolecular biofilm components yield EPS as a viscoelastic structure (Sutherland, 2001), their digestion causes a loss in EPS viscosity and thereby should reduce forces involved in the occlusion of POM. The effect is expected to grow with increasing enzyme activity until the whole EPS matrix is dispersed.

In the following, LF-SOC is interpreted as SOC from released POM, since the share of both adsorbed DOM and colloids on captured dry mass is considered to be negligible after SPT treatment. Furthermore, LF-SOC transferred from the sediment fraction to light fractions due to enzymatic treatment is also interpreted as POM, as in contrast mineral associated organic matter of the HF is not assumed to be extractable at the applied energies (Cerli et al., 2012).

In accordance with the model, measured oLF-SOC releases indicate a trend for increased POM release with increasing enzyme addition (Fig. 2). The E4 scenario shows that relative oLF-SOC release increased by 63% (5% of $C_\Sigma$) compared to E1 at 50 J ml$^{-1}$, but its release is similar to the mean of the other treatments at 0 J ml$^{-1}$, 100 J ml$^{-1}$ and 150 J ml$^{-1}$. Noticeable deviations of E1 and E4 from the control do not match the usual significance criteria ($p<0.05$). However, the increase of the relative oLF-SOC release in the E4 scenario compared to the control is predominantly related to an equally lower C content of the sediment but no decrease in the 100 J ml$^{-1}$ and 150 J ml$^{-1}$ fractions. That points to a strong (oLF >150 J ml$^{-1}$) intra-aggregate fixation of POM due to enzyme targets, which is weakened by enzymatic treatment.

The relation of LF-SOC release with enzymatic biofilm digestion is supported by the comparison of bacterial DNA releases between the treatments (Fig. 3). This indicates that

applied enzymes are targeting biofilm components and release bacterial cells: The E4 scenario shows EPS digestion and additional cell release leading to a doubled relative DNA release compared with the control and E1. However, considering that most of the soil

bacteria are expected to live in biofilms (Davey and O'toole, 2000), the total DNA release of only 5.6% in the E4 scenario is too low for total biofilm digestion. Hence, biofilm detachment caused by E4 is still likely to be incomplete and the increased oLF-SOC release of E4 only results from a partial soil biofilm detachment. We conclude a slight influence of enzymatic treatment on the occlusion of POM at enzyme concentrations

exceeding natural concentrations. This conforms to results of Böckelmann et al. (2003), which indicate that a treatment with enzyme concentrations of near that of E4 is sufficient to destabilize biofilms within 1 hour.

The incomplete biofilm detachment can be explained by the reduction of enzyme activity due to interaction with the soil matrix. Based on our calculations enzyme concentrations of

mix E1 should have been sufficient for total biofilm digestion within time of application (1h) – as far as there are no other factors reducing enzyme efficiency. As surveys of natural soils show enzyme concentrations up to mix E3 (Cooper and Morgan, 1981; Eivazi and Tabatabai, 1988; Margesin et al., 1999; Acosta-Martinez and Tabatabai, 2000; Margesin et al., 2000), such factors might be reasonably assumed. After addition to the soil sample,

enzymes must enter the EPS matrix by diffusion. Therefore parts of the enzymes probably do not reach the biofilm due to inhibited diffusion. Beside diffusion, sorption and decomposition could play a major role in reducing enzyme efficiency. Whereas turn-over rates of soil enzymes are not yet assessed, extended stabilization of active enzymes over time on soil mineral and organic surfaces is reported (Burns et al., 2013). This mechanism

could explain immobilization of enzymes off the biofilm and high measured soil enzyme concentrations from literature in face of still existing biofilms. After penetration of biofilms (macro)molecules interfere with EPS components depending on molecular size, charge and biofilm structure (Stewart, 1998; Lieleg and Ribbeck, 2011) which is strongly influencing decay rates of enzymes. Due to these boundary conditions, quantification of

the relation of enzyme concentration and POM carbon release was not possible in this work.

The trend for increased POM release with increasing enzyme addition was only broken by the control treatment. Whereas E4 matches the forecast of releasing more POM than the

control, scenario E1 shows a reduced release by -2.8% and the DNA release remains unchanged compared to the control. This decrease in the 50 J ml$^{-1}$ fraction is related to an increase in the sediment fraction and cannot be explained by the model (Fig. 1). Probably it could be explained by pre-incubation of soil aggregates given 0.2 mM $NH_4NO_3$ and further addition of $NH_4NO_3$ with enzyme application: Redmile-Gordon et al. (2015) proposed that low C/N ratios of substrates available to soil microorganisms reduce cell specific EPS production rates, and may trigger microbial consumption of EPS to acquire C for cell-growth, which could weaken the biofilm. The observations leading to this proposed dynamic were also found by addition of $NH_4NO_3$. In the present study, $NH_4NO_3$ was applied with all treatments including the control (which also received no C from enzyme provision). Enzyme C in E1 to E4 could be used as microbial C source. The addition of SOC is known to lead to soil aggregate stabilization (Watts et al., 2005; Tang et al., 2011) and withdraw the effect of reduced C/N ratio. In contrast, the retention of the lowest C/N ratio in the control soils may itself have sustained EPS consumption and repressed reconstruction of the EPS, contributing to the higher than expected release of POM from the control soil with sonication at 50 J mL$^{-1}$ and the break in the trend for increasing POM release with increasing enzyme addition. However, decay rates of enzymes in soil are unknown but needed for a more accurate estimation of enzyme C as a fast energy and carbon source.

Under certain conditions POM carbon release is indicative for soil aggregate stability. Generally, aggregate stability is characterized by determining the reduction in aggregate size after application of mechanical force. The commonly used methods are dry and wet sieving. However, the destruction of soil aggregates by ultrasonication has an advantage over these methods, which is the quantification of the applied energy (North, 1976). It is used for studying reduction of aggregate size (Imeson and Vis, 1984) as well as detachment of occluded POM carbon (Golchin et al., 1994). Kaiser and Berhe (2014) reviewed 15 studies using ultrasonication of soil aggregates in consideration of its destructiveness to the soil mineral matrix and occluded POM. They found destruction of POM at applied energy levels >60 J/ml, destruction of sand-sized primary particles at >710 J/ml and of smaller mineral particles at even higher energy levels. We used this method of gentle POM detachment from soil aggregates to measure the oLF-SOC release as a result of mechanical force and linked it to aggregate stability. Since Cerli et al. (2012) have shown that the release of free and occluded light fractions strongly depends on soil

properties like mineralogy, POM content, composition and distribution, this method is restricted to comparison of soils being similar in these properties. Having regard to this restriction, the trend for increase of oLF-SOC release over increasing enzyme additions demonstrates an alteration of soil aggregate stability.

Although our results give a slight evidence for the influence of biofilms on aggregate stability, they have to be recognized with restrictions to full quantifiability: (1) The enzyme concentration hypothetically needed to disperse the whole soil sample EPS matrix depends on diverse boundary conditions like the concentration of enzyme targets, environmental conditions such as pH, temperature as well as ion activity and delay factors

such as low diffusion, kinetic influence or metabolization of enzymes by soil organisms. (2) Underlying enzyme kinetics were measured by the producer using pure targets for unit definition, while biofilm targets are much more diverse and soil matrix could interfere. (3) Alternative enzyme targets might be reasonably assumed within the complex chemism of the soil matrix. Released organic cytoplasm molecules of lysed cells can be excluded to be

an additional enzyme target due to their low concentration. On the other hand, enzyme specificity to EPS targets in face of the organic soil matrix is unbeknown. (4) The decrease of extracted POM mass due to biofilm erasement from surfaces is suggested to be low, but could cause underestimation of POM release especially in scenario E4. In contrast, a direct contribution of enzyme C to the POM carbon release can be refused. Even in case

of complete adsorption to the POM of only one fraction, the highest enzyme concentration (E4) would result in additional 13.5 µg enzyme /g dry soil being <0.4% of the smallest extracted POM fraction (Table 3). (5) Regarding DNA release measurement as well, data are semi-quantitative, since quantification of the detachment effect is limited by a potential adherence of detached cells to soil particles after washing (Absolom et al., 1983; Li and

Logan, 2004). Thus, cell release could be underestimated as biofilm detachment increases.

Many of these uncertainties are owed to the high complexity of the soil system. Enzymes were applied in concentrations four orders of magnitude higher than calculated from actual $C_{mic}$ and even 1-2 orders of magnitude higher than values from literature. Incomplete

biofilm removal indicated by the release of maximum 5.5% DNA from the soil matrix may suggest that the pooled influence of the disregarded boundary conditions on enzymatic detachment efficiency is large.

However, these results give a first though still vague insight in fundamental processes underlying POM occlusion. A slight release of occluded POM coupled with increased bacterial DNA release after treatment with high enzyme concentrations underpin the assumption that biofilm is involved in POM occlusion being a stabilizing agent of soil aggregates as proposed in a review by Or et al. (2007). The apparent increase of POM carbon release caused by the digestion of EPS components suggests biofilm relevance in soil ecosystems e.g. in terms of soil-aggregate related functions like soil water and C dynamics, mechanical stability as well as rootability. However, the statistical power of this introductory work is low and a more quantitative analysis of the relation of enzymatic EPS detachment and POM release would require deeper knowledge of enzyme dynamics in soil, more replicate samples, additional enzyme concentrations and probably inclusion of soils from different land use. However, this was beyond the scope of the present study.

## 5 Conclusions

Extracellular polymeric substance (EPS) was shown to be a promising candidate factor of aggregate stability. Our experimental results suggest that EPS contributes to occlusion and attachment of particulate organic matter (POM) in sandy soil aggregates. The application of a highly concentrated mix of α-glucosidase, β-galactosidase, DNAse and lipase is related to a slight detachment of POM from a stable to a more fragile binding structure, but not to an increase in POM release without physical disruption of aggregates by sonication. The pattern of measured light fraction soil organic carbon (LF-SOC) release and additional bacterial DNA release points to an intra-aggregate fixation of POM by enzyme targets. A loss of EPS integrity could therefore cause a detachment of soil organic matter, not only in the laboratory but also in tilled soils. Our results further suggest that a change of the biofilm composition probably due to a shift in microbial population structure may alter soil aggregate stability. On macro-scale this could affect soil compactibility, erodibility, water transport, retention and aeration regime, rooting depth and the occlusion of soil organic carbon. This, in conclusion, invites to behold soil EPS dynamics as a factor of sustainable land use.

**Acknowledgements**

This project was financially supported by the Leibnitz-Gemeinschaft (SAW Pact for Research, SAW-2012-ATB-3). We also are grateful to Prof. Dr. Ulrich Szewzyk and the helpful staff of the Chair of Environmental Microbiology (Department of Environmental Technology, TU Berlin) for the unbureaucratic possibility to use their laboratories. In addition our thanks go out to Lara Schneider. Her bachelor thesis helped us to get an overview in the early phase of this experiment. And thanx both to Dennis Prieß for initiating a good idea during a very good Taiji training and to Tom Grassmann for his help to handle R coding.

**Author contribution**

The experiments were designed, carried out and data were evaluated by F. Büks. The manuscript was prepared by F. Büks with contributions from M. Kaupenjohann.

## Data availability

**Absolom, D. R. et al.**,  1983, URL: http://aem.asm.org/content/46/1/90.short

**Acosta-Martinez, V. and Tabatabai, M.**, 2000, doi:10.1007/s003740050628

**Agnelli, A. et al.**,  2004, doi:10.1016/j.soilbio.2004.02.004                    550

**Alaoui, A. et al.**,  2011, doi:10.1016/j.still.2011.06.002

**Ashman, M. et al.**,  2009, doi:doi:10.1016/j.still.2008.07.005

**Baldock, J.**, 2002, ISBN: 0-471-60790-8

**Ball, B. and Robertson, E.**,  1994, doi:10.1016/0167-1987(94)90076-0

**Barthes, B. and Roose, E.**,  2002, doi:10.1016/S0341-8162(01)00180-1            555

**Battin, T. J. et al.**,  2007, doi:doi:10.1038/nrmicro1556

**Baumgartl, T. and Horn, R.**,  1991, doi:10.1016/0167-1987(91)90088-F

**Bengough, A. and Mullins, C.**,  1990, URL:
http://onlinelibrary.wiley.com/doi/10.1111/j.1365-2389.1990.tb00070.x/pdf

**Bennie, A. and Burger, R. d. T.**,  1988, doi:10.1080/02571862.1988.10634239    560

**Böckelmann, U. et al.**,  2003, doi:10.1016/S0167-7012(03)00144-1

**Bratbak, G. and Dundas, I.**,  1984, URL: http://aem.asm.org/content/48/4/755.short

**Brauns, A.**, 1968

**Brodowski, S. et al.**,  2006, doi:10.1111/j.1365-2389.2006.00807.x

**Bronick, C. J. and Lal, R.**,  2005, doi: 10.1016/j.geoderma.2004.03.005          565

**Brown, G. G. et al.**,  2000, doi:10.1016/S1164-5563(00)01062-1

**Burns, R. G. et al.,** 2013, doi: 10.1016/j.soilbio.2012.11.009

**Cerli, C. et al.**,  2012, doi:10.1016/j.geoderma.2011.10.009

**Chaudhari, P. R.** et al., 2013, ISSN: 2250-3153

**Chenu, C. and Stotzky, G.**, 2002, ISBN: 0471607908                              570

**Chenu, C. et al.**,  2002, ISBN: 978-0-471-60790-8

**Cooper, A. and Morgan, H.**, 1981, doi:10.1016/0038-0717(81)90067-5

**Das, T. et al.**,  2014, doi:10.1371/journal.pone.0091935

**Davey, M. E. and O'toole, G. A.**,  2000, doi:10.1128/MMBR.64.4.847-867.20

**DeFlaun, M. F. et al.**,  1987, URL: http://scholarcommons.usf.edu/cgi/viewcontent.cgi?    575
article=1101&context=msc_facpub

**Eivazi, F. and Tabatabai, M.,** 1988, doi:10.1016/0038-0717(88)90141-1

**Eusterhues, K. et al.**, 2008, doi: 10.1021/es800881w

**Flemming, H.-C. and Wingender, J.**,  2010, doi:10.1038/nrmicro2415

**Golchin, A. et al.,** 1994, doi: 10.1071/SR9940285                               580

**Imeson, A. and Vis, M.**, 1984, doi: 10.1016/0016-7061(84)90038-7

**Jahn, A. et al.**,  1999, doi:10.1080/08927019909378396

**Joergensen, R. G.**,  1996, doi:10.1016/0038-0717(95)00102-6

**Kaiser, M. and Berhe, A. A.**,  2014, doi:10.1002/jpln.201300339

**Kalbitz, K. et al.**, 2000, URL:                                                 585
http://journals.lww.com/soilsci/Abstract/2000/04000/Controls_on_the_Dynamics_of_Dissol
ved_Organic.1.aspx

Lal, R., 2008, doi: 10.1039/B809492F

Lehmann, J. and Kleber, M., 2015, doi: 10.1038/nature16069

Li, B. and Logan, B. E., 2004, doi:10.1016/j.colsurfb.2004.05.006

Lieleg, O. and Ribbeck, K., 2011, doi:10.1016/j.tcb.2011.06.002

Lützow, M. v. et al., 2006, doi:10.1111/j.1365-2389.2006.00809.x

Marchesi, J. R. et al., 1998, URL: http://aem.asm.org/content/64/2/795.short

Margesin, R. et al., 1999, doi:10.1023/A:1008928308695

Margesin, R. et al., 2000, doi:10.1016/S0045-6535(99)00218-0

More, T. et al., 2014, doi:10.1016/j.jenvman.2014.05.010

Munk, K., 2008, ISBN: 978-3-13-144831-6

Muyzer, G. et al., 1993, URL: http://aem.asm.org/content/59/3/695.short

Niemeyer, J. and Gessler, F., 2002, URL: http://onlinelibrary.wiley.com/doi/10.1002/1522-
2624%28200204%29165:2%3C121::AID-JPLN1111121%3E3.0.CO;2-X/pdf

North, P., 1976, doi: 10.1111/j.1365-2389.1976.tb02014.x

Nunan, N. et al., 2003, doi: 10.1016/S0168-6496(03)00027-8

Oades, J., 1993, doi:10.1016/0016-7061(93)90123-3

Or, D. et al., 2007, doi: 10.1016/j.advwatres.2006.05.025

Pal, A. and Paul, A., 2008, doi:10.1007/s12088-008-0006-5

Pietramellara, G. et al., 2009, doi:10.1007/s00374-008-0345-8

Pokrovsky, O. S., 2005, doi: 10.1007/s10498-004-4765-2

Redmile-Gordon, M. et al., 2014, doi:10.1016/j.soilbio.2014.01.025

Redmile-Gordon, M. et al., 2015, doi: 10.1016/j.soilbio.2015.05.025

Rillig, M. C., 2004, doi:10.4141/S04-003

Schmidt, M. et al., 1999, URL: http://onlinelibrary.wiley.com/doi/10.1046/j.1365-
2389.1999.00211.x/pdf

Schmitt, J. and Flemming, H.-C., 1999, doi:10.1016/S0273-1223(99)00153-5

Six, J. et al., 2002, URL: http://link.springer.com/article/10.1023/A:1016125726789

Six, J. et al., 2004, doi: 10.1016/j.still.2004.03.008

Skidmore, E. and Powers, D., 1982, doi:10.2136/sssaj1982.03615995004600060031x

Stewart, P. S., 1998, doi:10.1002/(SICI)1097-0290(19980805)59:3<261::AID-
BIT1>3.0.CO;2-9

Sutherland, I. W., 2001, doi:10.1099/00221287-147-1-3

Tang, J. et al., 2011, doi: 10.1016/j.apsoil.2011.01.001

Taylor, H. and Brar, G., 1991, doi:10.1016/0167-1987(91)90080-H

Tisdall, J., 1991, doi:10.1071/SR9910729

Tisdall, J. and Oades, J., 1982, doi:10.1111/j.1365-2389.1982.tb01755.x

Wagai, R. et al., 2009, doi: 10.1111/j.1747-0765.2008.00356.x

Walker, S. L. et al., 2007, doi:10.1002/j.2050-0416.2007.tb00257.x

Weng, L. et al., 2002, doi: 10.1021/es0200084

Wright, S. and Upadhyaya, A., 1998, doi:10.1023/A:1004347701584

Zhang, X. et al., 1998, doi:10.1016/S0273-1223(98)00127-9

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

*Tables*

**Table 1:** Concentrations and molar masses of biofilm stabilizing macromolecules (polysaccharides=PS, eDNA, lipids and proteins) in different environments.

| Conc. µg (g soil)$^{-1}$ | Proportion µg (100 µg EPS)$^{-1}$ | Molar mass Da | Comment | Reference |
|---|---|---|---|---|
| *PS* | | | | |
| 169 | | | µg g$^{-1}$ bare fallow | [Redmile-Gordon et al., 2014] |
| 401 | | | µg g$^{-1}$ grassland | [Redmile-Gordon et al., 2014] |
| | 95 % | | % of EPS dry-mass | [More et al., 2014] |
| | 40-95 % | | % of EPS dry-mass | [Pal and Paul, 2008] |
| | | 2x10$^6$ | | [Chenu and Roberson, 1996] |
| | | 0.5-2x10$^6$ | | [Flemming and Wingender, 2010] |
| *eDNA* | | | | |
| 2.2-41.1 | | | µg g$^{-1}$ forest soil | [Agnelli et al., 2004] |
| 0.08 | | | µg g$^{-1}$ Luvisol | [Niemeyer and Gessler, 2002] |
| 1.95 | | | µg g$^{-1}$ forest podzol | [Niemeyer and Gessler, 2002] |
| 0.03-200 | | | µg g$^{-1}$ unnamed soil | [Pietramellara et al., 2009] |
| | 10 % | | % EPS dry-mass | [More et al., 2014] |
| | | 7.75x10$^4$-2.32x10$^7$ | estuarine and oceanic environments | [DeFlaun et al., 1987] |
| *Lipids* | | | | |
| | 10 % | | % of EPS dry-mass | [More et al., 2014] |
| | | 750-1500 | | [Abröll and Munk, 2008] |
| *Proteins* | | | | |
| 43 | | | µg g$^{-1}$ bare fallow | [Redmile-Gordon et al., 2014] |
| 163 | | | µg g$^{-1}$ grassland | [Redmile-Gordon et al., 2014] |
| | < 75 % | | % of *Ps. Putida* biofilm | [Griebe and Nielson, 2000] |
| | 60 % | | % EPS dry-mass | [More et al., 2014] |



**Table 2:** Variables used for the calculation of enzyme Units needed for biofilm target decomposition and scenario parameters shared by all variants, [a] More et al., 2014; [b] Pal and Paul, 2008; [c] Flemming and Wingender, 2010; [d] Abröll and Munk, 2008; [e] DeFlaun et al., 1987.

| | | |
|---|---|---|
| $c_{cell}$ | [µg g$^{-1}$] | bacterial dry mass per g dry soil |
| $q$ | [-] | enzyme concentration multiplier |
| $\xi_{EPS}^{max}$ | [-] | maximum ratio of EPS dry mass per total biofilm dry mass ( $\xi_{EPS}^{max} = 0.9^{[a]}$ ) |
| $\xi_{target}^{max}$ | [-] | maximum ratio of enzyme target per EPS dry mass ( $\xi_{polysaccharides}^{max} = 0.95^{[b]}$ , $\xi_{lipids}^{max} = 0.1^{[a]}$ and $\xi_{eDNA}^{max} = 0.1^{[a]}$ ) |
| $m_{sample}$ | [g] | sample mass |
| $\xi_{cell}^{min}$ | [-] | minimum ratio of bacterial dry mass per total biofilm dry mass ( $\xi_{cell}^{min} = 0.1^{[a]}$ ) |
| $M_{target}^{min}$ | [µg µmol$^{-1}$] | minimum molar mass of enzyme target ( $M_{polysaccharides}^{min} = 0.5 \, x \, 10^{6[c]}$ , $M_{polysaccharides}^{min} = 700^{[d]}$ , $M_{eDNA}^{min} = 7.75 \, x \, 10^{4[e]}$ ) |
| $t$ | [min] | incubation time |





**Table 3:** Specific scenario parameters of the variants E0, E1, E2, E3 and E4.

| | | E0 | E1 | E2 | E3 | E4 |
|---|---|---|---|---|---|---|
| $c_{cell}$ | [µg g$^{-1}$ dry soil] | 352 | 352 | 1191 | 1191 | 1191 |
| $q$ | [-] | 1 | 1 | 1 | 100 | 2,820 |
| $U_{a-glucosidase}^{max}$ | [U g$^{-1}$ dry soil] | 0.00000 | 0.00010 | 0.00034 | 0.03393 | 0.95683 |
| | [µg g$^{-1}$ dry soil] | 0.00000 | 0.00080 | 0.00272 | 0.27144 | 7.65464 |
| $U_{b-galactosidases}^{max}$ | [U g$^{-1}$ dry soil] | 0.00000 | 0.00010 | 0.00034 | 0.03393 | 0.95683 |
| | [µg g$^{-1}$ dry soil] | 0.00000 | 0.00020 | 0.00068 | 0.06786 | 1.91366 |
| $U_{lipids}^{max}$ | [U g$^{-1}$ dry soil] | 0.00000 | 0.00754 | 0.02551 | 2.55102 | 71.93876 |
| | [µg g$^{-1}$ dry soil] | 0.00000 | 0.00038 | 0.00126 | 0.12551 | 3.59694 |
| $U_{eDNA}^{max}$ | [U g$^{-1}$ dry soil] | 0.00000 | 0.00007 | 0.00023 | 0.02304 | 0.64973 |
| | [µg g$^{-1}$ dry soil] | 0.00000 | 0.00004 | 0.00012 | 0.01152 | 0.32487 |




*Figures*

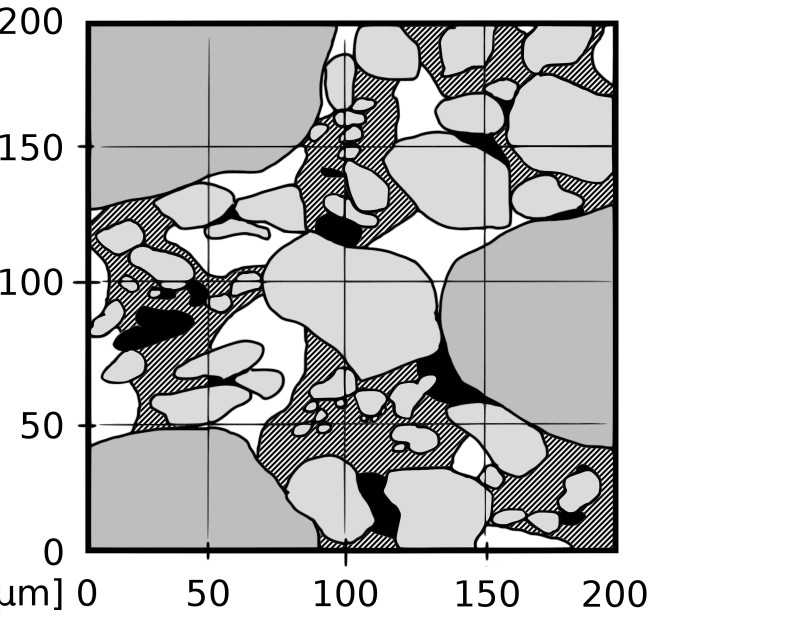

**Fig. 1:** Proposed model of aggregate structure including biofilms in a soil aggregate: Sand and silt (both grey) and organic particles (black) stick together by physico-chemical interactions and are bridged by EPS (striped), which additionally stabilizes the soil aggregate structure and the pore space (white).

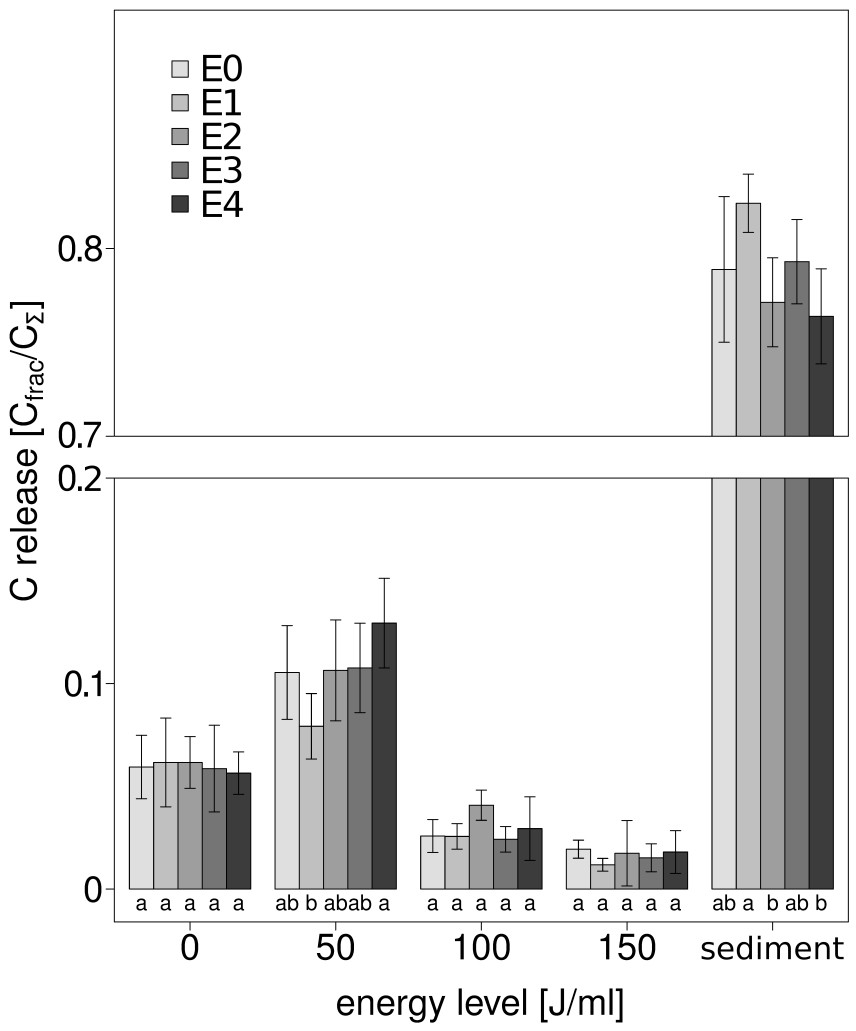

**Fig. 2:** Relative POM carbon release of treatments (E0, E1, E2, E3, E4) at different energy levels (0, 50, 100, 150 J ml$^{-1}$, sediment), illustrated by Tukey test characters (a, ab, b). Data are shown as mean values and standard deviations (n=5).

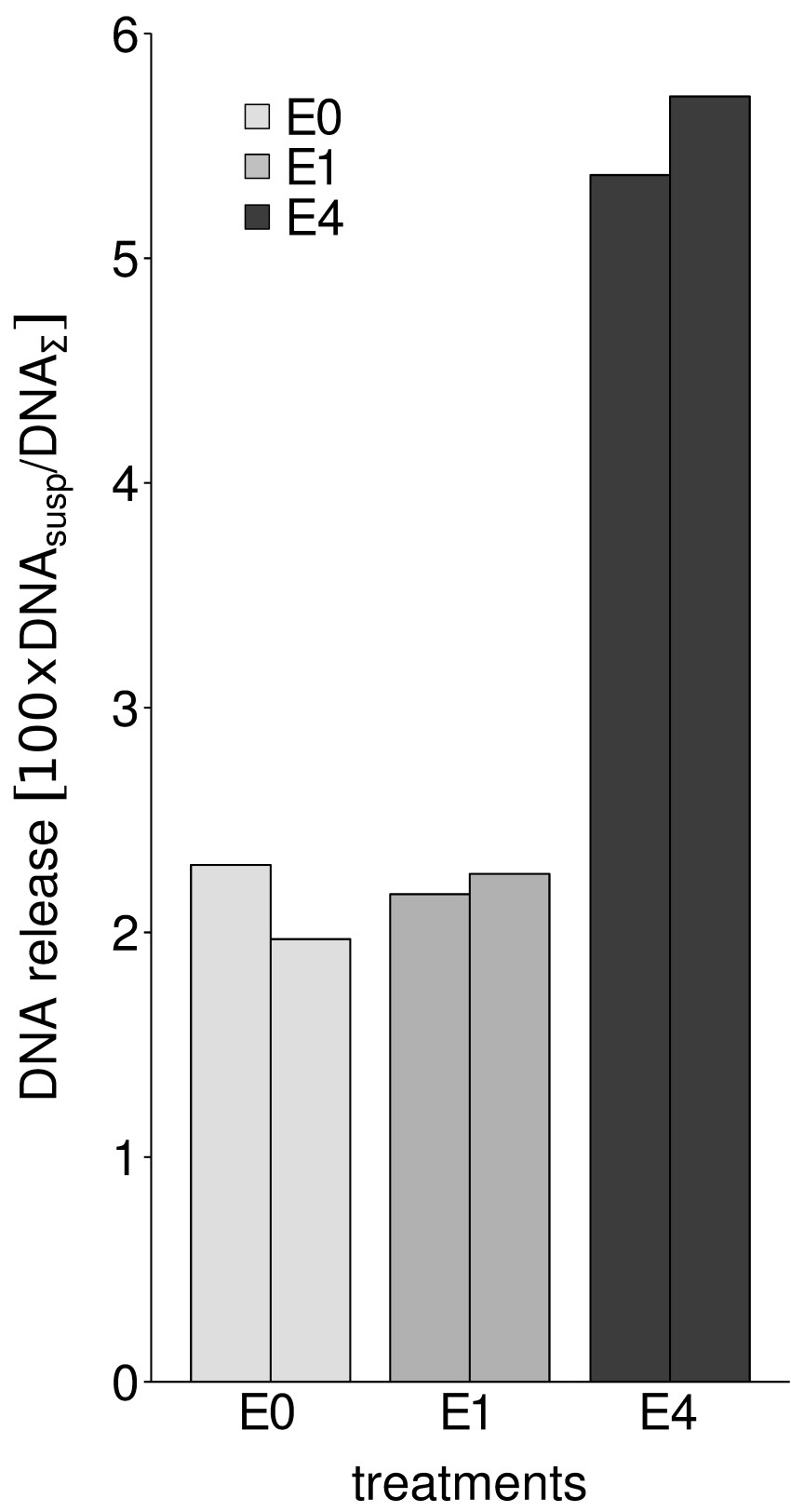

**Fig. 3:** Relative bacterial DNA release from soil aggregates after treatments E0, E1, and E4 defined as 100x ratio of bacterial DNA from suspended cells ($DNA_{susp}$) to total bacterial DNA from suspended cells, sessile cells ($DNA_{\Sigma}$) and the EPS remaining upon the soil matrix.