# Peer review of "Enzymatic biofilm digestion in soil aggregates facilitates the release of particulate organic matter by sonication"

_SOIL, 2015_

## Referee Comment (RC1) · 6 Mar 2016

A fully formatted document is attached as a PDF for reference.

Manuscript review of soil-2015-87 (Submitted on 03 Dec 2015) Enzymatic biofilm detachment causes a loss of aggregate stability in a sandy soil Authors: F. Büks and M. Kaupenjohann

Non-anonymous review offered by: Marc Redmile-Gordon (Rothamsted Research, UK)

General comments This paper presents an interesting and novel approach to cause biofilm detachment from soil minerals using enzyme digestion, followed by ultrasonic disaggregation and measurement of the particulate organic matter (POM) subsequently released. It is multidisciplinary and of broad international interest within soil

and biofilm disciplines. The work represents a substantial contribution to scientific progress and fits comfortably within the scope of SOIL.

However, the manuscript is currently poorly presented and needs care. Over-emphasis and repetition of non-significant results and poor discussion detract from the valued approach and findings. Before I am able to recommend publication, there are numerous statements, assumptions and conclusions that I think need reconsideration. The discussion in particular needs substantial improvement on the version currently available online (1st April 2016). I have made several suggestions and would encourage the authors to attend to this at their earliest convenience for publication of this important manuscript.

I would be willing to conduct a repeat review of the revised version, and hope my comments are found useful.

Specific comments While most scientific questions/issues are given in line order under the 'technical corrections' section, I would initially draw attention to the following:

1 - Title: Enzymatic biofilm detachment causes a loss of aggregate stability in a sandy soil While this statement is probably true, the work done is not actually sufficient premise for the title. The release of POM is likely to be associated with greater disruption of aggregates, but, as I can see, aggregate stability per se was not measured directly, and the release of POM may have been caused by another factor(s). Since aggregate stability may or may not have been affected I suggest a more conservative replacement title: 'Enzymatic biofilm digestion in soil aggregates facilitates the release of particulate organic matter (POM) by sonication'. I think this title would also be more informative, useful, and accurate.

Accordingly, at line 264 I suggest heading renamed to 'Intra-aggregate POM' because 'Aggregate stability' suggests physical stabilities of aggregate size classes (not measured). There are many methods to quantify 'aggregate stability', none of which are described or referenced in the methods of the manuscript. This is one example of

rather too many leaps or assumptions at present that need taming.

There is circular overdescription of results. For example, at lines 284-288, contrary to the images of 'shift' and 'transfer' there has been no physical transfer of organic matter from the oLF fraction to the sediment. There were instantaneous observations of POM in different fractions that do not exist until one creates them in the laboratory. The different quantities of POM released simply indicate the effects of the enzyme treatment (and elegantly so). This may be due to a reduced aggregate stability for example, and/or additionally increased fragmentation of POM, possibly including biofilm fragments (including cells and EPS) from the soil that are large enough (or sticky enough) to be retained on the 1.5 $\mu$M glass filter. This is a matter perhaps for the discussion. Nonetheless, the model proposed to explain the observed phenomenon (of increased POM release with enzyme addition and sonication) is one of reduced aggregate stability by EPS removal – I think this is reasonable and should be retained (but not stated as fact without harder evidence).

Further comments and general technical corrections

1 Title should change, I suggest: 'Enzymatic biofilm digestion in soil aggregates facilitates the release of particulate organic matter (POM) by sonication'. 13 please choose between either: 'shear and compression' or 'shearing and compressive'. 14 replace 'high stability results in less soil compatibility and erodability' with 'high stability results in less compaction and erosion, and has been linked to '. 15 delete 'a' in 'a dynamic'; replace 'regime' with 'regimes'. 24 delete 'which preserves aggregate structure': I do not think is correct to say this, physical aggregate structure was not measured directly; it was inferred indirectly via release of POM-C. Furthermore, some structure must have been changed enzymatically to result in the increased release of C. To avoid philosophical digression into definitions of 'biological', 'physical', and 'chemical' structure in the abstract, the manuscript is better without it: just delete. 26 replace 'effectivity' with 'efficacy' 30 This is overly confident and not quite accurate. Is it not true that enzymatic digestion of EPS polymers may have increased the abundance

of EPS fragments released upon sonication? Therefore, remove 'our results confirm, that EPS stabilises soil aggregates predominantly by a strong intra-aggregate fixation, and enzymatic biofilm digestion caused a shift of occluded particulate organic matter (POM) to more fragile binding patterns' and replace with 'our results suggest that EPS stabilises intra-aggregate particulate organic matter (POM) within soil aggregates'.

Introduction 37 replace compressive with compression 38 replace 'water caused lability' with 'disaggregation caused by water' 38 insert 'and' before 'is an integral' 39 replace 'Because' with 'Since' 46 delete 'the' from 'the water retention' 46 insert a comma after 'furthermore' 49 replace 'SOM promotes' with 'SOM is thought to promote' 56 delete '.' 61 – 63 awkward sentence, please rephrase 78 check spelling polysaccharide 79 replace 'exudated' with 'exuded' 80 insert 'being' before engineered 82 replace 'biofilm forming species and habitats:' with 'community composition and environmental cues:' 83 replace 'determined' with 'measured' 89 reverse word order, should be 'other diverse soils ranges between' 92 replace 'can amount to' with 'was measured at' 93 replace 'and often hold 60% of EPS dry mass' with 'but can contribute the largest fraction of EPS, e.g. 60%' 96 scientific notation check: I suspect this should be 0.5 x 106 (and likewise throughout the rest of the manuscript). 101 'ranges' 105 should eDNA be eDNAase? 108 Unsubstantiated statement which leaves the reader wondering 'why'. I suspect the authors are drawing on the rationale presented Redmile-Gordon et al. (2014) and suggest this is expanded upon for clarity and to help build justification. Suggest the authors replace 'That is mainly due to methodological reasons' with 'This is mainly due to methodological reasons. For example, Tang et al. (2011) found no link between bacterial EPS extracted using sulphuric acid and aggregate stability. Redmile Gordon et al (2014) subsequently found in a comparison study that the techniques previously used to measure extracellular polysaccharide in soil co-extracted large quantities of 'random' soil organic matter which confounded estimates of EPS production." 110 replace 'hardly effecting other aggregate binding mechanisms' with 'with minimum impact on other types of aggregate bond' 112 replace 'coherence' with 'cohesion' 112 remove 'and aggregate stability' (I do not think you measured this)

consider adding a statement like 'which is expected to contribute to aggregate stability' (then reference needed). 123 replace 'after' with 'occurs during' 123 replace 'that' with 'this'

Materials and methods 138 replace 'receive' with 'obtain' 141-142 This is not a method to estimate soil microbial biomass, this is respiration, correct accordingly. 143 delete 'then' 144 replace 'have been' with 'were' 146 careful with pluralities, use 's' and 'were' 147 $\mu$g g-1 is more appropriate 172 This section takes some time to understand. Insert "sufficient enzymes were provided to digest the EPS content expected in five scenarios (E0 to E4)" 173 use large 'C' for carbon 181 Renamed to 'Intra-aggregate POM' (or similar) no method or results for a measurement of aggregate stability are presented or referenced: e.g. Cerli et al 2012 do not claim this method quantifies aggregate stability. 182 Accordingly, delete 'soil aggregate stability was measured at the macro scale'. 190 Why for 30 min? To allow NaPT diffusion? State your reason. 192 Unlike Cerli et al (2012), it is currently a little ambiguous if this fLF passed the filter or was captured on the filter. I expect the latter. Suggest you replace 'separate' with 'capture'. 195 50 J ml-1 given over what time period? 200 Total = sum of C in the sediment and filters? Be explicit (you did not quantify DOC, correct?) 210 'washed three times' 217 what volume of wash was used as an equivalent for the mass of soil stipulated in the FastDNATM spin kit soil manual? (Can it really be used to extract DNA from a dilute wash and compare with soil?).

RESULTS 263 rename section 3.1 as method above 264 delete 'the relative net SOC release shows' (you did not quantify DOC which is implied in 'SOC'). 265 insert 'are shown' before '(Fig. 1)' 265 replace SOC with POM 266 replace 'organic carbon' with 'POM' 268 move 'data are shown as mean values and standard deviations of five parallels' to figure caption. 270 replace SOC with POM 272 replace SOC with POM 273 Replace 'SOC of the FLF' with 'quantity of POM released in the absence of sonication' 274-276 Incorrect (and potentially misleading) presentation of results. Suggest as replacement: "there was no increase or decrease relative to the control, however,

there was a trend for increased POM release with increasing enzyme addition, and the difference between the lowest enzyme addition and the highest was statistically significant as indicated by the Tukey test. This trend was only broken by the control treatment (given no enzymes)" 276-277 Unnecessary and confusing statement, we can see the standard deviation and Tukey test results on the figure, better to remove the statement. 277-279 Potentially misleading statement, yes, E2 and E3 have no difference compared to the control, but neither do E1 or E4. See comment above re 274-276. 280 Replace SOC with POM. Delete 'stock'. 'varying between variants'? Do you mean 'different among treatments'? 281 replace 'an in' with 'any' 283 replace 'is therefore differing between variants with' with 'therefore shows' 284-288 There has been no physical transfer of organic matter between these analytical pools. A reduced aggregate stability may have for example, or increased release of biofilm fragments retained on the 1.5 $\mu$M glass filter, but this is a matter for discussion. . . . . .It might be more useful to say here that it is reassuring that the SOC remaining in the sediment reflects what would be expected given the quantities extracted at 50 J... but of course it would (because you present relative fractions in preference to absolute concentrations). I am struggling to find a reason to retain this section. I think it better to delete lines 284-288. 289 and Figure 2 These results have already been presented, it is not clear exactly what compounded estimate of error is being given, and besides, data were already presented in figure 1. Remove Figure 2. Delete line 289. 290 This has already been presented, that one can add the non-significant results to the significant, and finds the same thing is nothing surprising or worthy of comment. Delete. 291 Clumsy sentence and repetition: delete first sentence. 291-293 Released POM data may be evidence of this, and may not be - this is a matter for the discussion. Delete these lines. 293-296 Delete section starting "The lower aggregate stability is indicated by a steeper gradient and on average in an. . .". Replace with "The addition of the highest enzyme concentration (E4) caused the release of about 40% more POM by mild sonication (50J ml-1) than occurred with the addition of the lowest concentration (E1). This was statistically significant at (p <0.05)." end of section. 302 In contrast here I

think the relative increase in DNA release is a little understated. Yes it is useful to also give it as a percentage of total DNA extracted from the soil as you have done (Figure 3 - now rename to Figure 2), but perhaps in line 302 replace text "it is increased by about 3.5% to a value of 5.5% in the E4 scenario in comparison to the control" with 'While there was no difference in DNA concentrations suspended in the wash of control and low enzyme additions, treatment E4 caused an increase to more than double the DNA content of either E0 or E1."

DISCUSSION 324-335 First paragraph disorganised: it is an unpleasant jump to the model in the first sentence. Build up to it. It would be smoother if begin with the main result result, followed by your description of enzyme transport into the unsaturated pore space and discussion of others work E.g. "We found that increasing the quantity of enzymes applied to aggregates led to increased release of POM when aggregates were sonicated. Then describe the pore system (currently lines 325 326), then give your model of explanation "we present a model to explain the observed findings. . ." 336-337 Delete the discussion of what is not being discussed. 339 Replace SOC with POM 340 Replace SOC with POM 342 Remove paragraph break 343 Replace 'SOC' with 'POM'; replace 'and' with 'with'; Replace 'enforced' with 'supported' 344 Full stop after (Fig. 3). replace 'which' with 'This'. 345 'de facto' is way too strong and encourages the reader think of examples to disprove this over-confident statement. E.g. it could have been caused by cell lysis. Delete 'de facto'. 352 – This is not the only possible explanation and further discussion with relevant literature is required. Might some of the C released from occluded POM and/or biofilm not have been detected in the filtered light fraction? – e.g. may have been present as smaller particulates or DOC? Also, DNA/cells/POM may not have been released without sonication. Include this. Current literature has more to offer. Add "Furthermore, we pre-incubated soils given 0.2 mM NH4NO3, and added further NH4NO3 with the enzyme application. Redmile-Gordon et al (2015) proposed that low C/N ratios of substrates available to soil microorganisms reduces cell-specific EPS production rates, and may trigger microbial consumption of EPS to acquire C for cell-growth. The observations leading to this proposed dynamic

were also found by addition of NH4NO3. In the present study, NH4NO3 was applied with all treatments including the control (which also received no C from enzyme provision). The resulting lowest C/N ratio in the control soils may itself have decreased the EPS, contributing to the higher than expected release of POM from the control soil with sonication at 50 J mL-1, and the break in the trend for increasing POM release with increasing enzyme addition." 353 – Replace "The incomplete. . . . . . .ambiguously" sentence with "Nonetheless, biofilm detachment caused by E4 is still likely to be incomplete." And continue with "Slow enzyme diffusion. . ." 356 - 367 This paragraph contains some useful information that should be retained for comparison of enzyme quantities added. However, the explanation drawing on enzyme activities in natural soils is not clear and needs re-thinking and re-writing. Actually, it seems the argument is flawed. You only observed effects when you increased enzyme activities well above 'natural' levels so on the contrary seems to support the hypothesis that diffusion factors ARE limiting (e.g. sorption to active surfaces). Suggest you cite the excellent review by (Burns et al., 2013) (see section 3.3; page 220). 368 – 370 It does not reinforce this, and if it does it conflicts with your model. If your model is correct it would only be found after disruption of aggregates to release the oLF (as you observed at 50 J ml-1; congruent with your model).. It could also have been lost as soluble C, as mentioned above in reference to line 352 above. Delete 368 – 370. 371 & 379 - Replace SOC with POM 378 - 383 Not statistically significant therefore remove this speculation. Statistically it is built on observations that can be reasonably expected by chance. 380 – There is no 'transfer' other than in abstract operationally defined concepts - delete statement. 384 – replace 'cumulation of LF carbon release overall energy level clarifies the alteration of soil aggregate stability' with 'The trend for increased of LF carbon release over increasing enzyme additions demonstrates an alteration of soil aggregate stability'. 385 – results repetition. 386 - 389 Careful, you are discussing SOC (POM) release and aggregate stability as if you measured both independently, and focus drifts. I recommend you instead discuss the connection you propose (POM release being due to digestion of EPS which seems to prevent POM release by sonication alone up to 150 J ml-1

– and after more effectively separated from soil minerals by 50J sonication). 395 – Good point re enzyme metabolism, although 1 hour is not a lot of time for it, it would be useful to include a reference for rapid metabolism of enzymes/proteins. Add that the large additions of enzyme-C could be used as a C-source for microbial growth which is known to stabilise soil aggregates, e.g. (Watts et al., 2005). This is why total enzyme-C added should be included in your manuscript (suggest this is added to Table 3). 406 – Replace 'most of this restrictions' with 'many of these uncertainties' 407 – better if you delete 'a 9000 fold of the E1 enzyme activity calculated from actual soil biomass to remove approximately' 408 – suggest replace '5.5% of the biofilm and no increase in FLF release, the pooled influence of the disregarded boundary conditions on enzymatic detachment efficiency is large' with '5.5% biofilm removal indicated by DNA measurements coupled with no increase in fLF release, may suggest that the pooled influence of the disregarded boundary conditions on enzymatic detachment efficiency is large' 409 - Insert sentence: 'Conversely, or in addition to the above, complete biofilm removal may have been achieved, however as the model (figure 4 – now figure 3) proposes, POM would not be released until the retaining aggregates were disrupted by disruptive physical forces such as those caused by sonication.' (Add Kaiser and Berhe reference 2014) 410 - delete 'nonetheless' 411 - replace 'Loss of aggregate stability' with 'Release of entrapped POM' 413 - replace 'stabilisation' with 'stabilising' 413 – Citation needed: suggest after 'stabilising agent of soil aggregates' to insert 'as discussed in a comprehensive review by Or et al. (2007)'. 413 – Subsequent sentence, why limit to just natural ones? I suggest you replace 'Aggregate stability is influenced by the digestion of EPS components. Adapting this relation to natural soil ecosystems,"' with 'The apparent loss of aggregate stability caused by the digestion of EPS components in the present study suggests biofilm relevance in soil ecosystems.' And finish the discussion there. 414 – 417 Move this final part to the start of conclusions: "Our results suggest a change of biofilm composition due to a shift. . ."

Conclusions Much here currently appears needless repetition of the results already discussed. 419 – 420 Already discussed, is weak, better to delete. 422 – 423 delete

"and thereby enhances aggregate stability". Already discussed and now superseded by your two important sentences above this (first one suggested to be taken from discussion, lines 414 – 417). 425 Delete 'fLF' (these abstract technical distinctions are not appropriate for this statement). Continue with the condition i.e. "not to an increase in fLF release without physical disruption of aggregates by sonication." 425-427 replace SOC with POM (should already be defined) 427 delete the sentence starting "The bacterial DNA…" as discussed already; this does not withstand logical critique. 431 'microbial communities' already are for various reasons, I think you mean the biofilm or EPS, EPS being relevant even when no biofilm can be observed… suggest you replace 'communities' with 'EPS dynamics'. Figures and Tables Renumber figures after deleting Figure 2 Figure 1: Replace SOC with POM Figure 3: Check scientific notation (Y axis) and replace '.' With 'x' ? Figure 4: edit caption – you are not showing 'biofilm structure' – this is 'aggregate structure' replace accordingly. Table 1: Check scientific notation under column 'Molar mass' ( e.g. should be 2 x 106 ?) Table 3: Add quantity of enzyme-C added to enable judgement of substrate utilisation by soil microbial biomass. Table 3: column E0: should the q value not be zero? Otherwise why are the enzyme activities different from column E1?

References Burns, R.G., DeForest, J.L., Marxsen, J., Sinsabaugh, R.L., Stromberger, M.E., Wallenstein, M.D., Weintraub, M.N., Zoppini, A., 2013. Soil enzymes in a changing environment: Current knowledge and future directions. Soil Biology & Biochemistry 58, 216-234. Or, D., Smets, B.F., Wraith, J.M., Dechesne, A., Friedman, S.P., 2007. Physical constraints affecting bacterial habitats and activity in unsaturated porous media - a review. Advances in Water Resources 30, 1505-1527. Redmile-Gordon, M.A., Evershed, R.P., Hirsch, P.R., White, R.P., Goulding, K.W.T., 2015. Soil organic matter and the extracellular microbial matrix show contrasting responses to c and n availability. Soil Biology and Biochemistry 88, 257-267. Tang, J., Mo, Y., Zhang, J., Zhang, R., 2011. Influence of biological aggregating agents associated with microbial population on soil aggregate stability. Applied Soil Ecology 47, 153-159. Watts, C.W., Whalley, W.R., Brookes, P.C., Devonshire, B.J., Whitmore, A.P., 2005. Biological

and physical processes that mediate micro-aggregation of clays. Soil Science 170, 573-583.

Please also note the supplement to this comment:
http://www.soil-discuss.net/soil-2015-87/soil-2015-87-RC1-supplement.pdf

---

## Referee Comment (RC2) · Anonymous Referee #2 · 22 Mar 2016

This manuscript (soil-2015-87) aims to provide a better understanding of the role of extracellular biological materials (EPS), esp found in biofilms, in soil aggregation stabilisation. It uses a series of additions of hydrolytic degradative enzyme to test the hypothesis that EPS materials contribute to the stability of soil aggregation, while also affecting SOM availability.

I have major concerns regarding this manuscript:

1. Poor study design and poor description of the methodologies used:
    a. Section 2.2: confusingly written maths section
    b. Section 2.2: poor justification of numbers used:
    Eg the supposed soil bulk density number seems odd, as this can be measured for field core samples and be recreated to field soil density. Otherwise explain the assumption for this particular experiment as normal dried and sieved soil without repacking does not get to this density.
    c. Section 2.2/2.3: poor justification of numbers used:
    The 'scenarios' have been explained (though could be improved in clarity) but do not actually contain any information regarding the technical set up. How much enzyme activity units were applied? What was the level of purity of the enzyme preparations? How where the enzymes added? Was there mixing involved? There is a severe lack of information, especially as the whole manuscript depends on contact of these enzymes with EPS materials. How have the authors assured that these enzymes have reached the materials processed further?
    d. Section 2.2/2.3: the E4 scenario seems to suggest a large excess of enzymes was applied. How have the authors ensured that such a large excess is not damaging to resident live microbial cells? E.g. a large excess of lipase may affect the membrane integrity of cells. This may in turn impact on DNA quantification without actually directly affecting soil aggregate stability.
    e. Section 2.3: information/studies on basal respiration at 30C/37C, the temperature of the actual experiments performed, are missing.
    f. Section 2.4: this experiment was performed on a separate soil incubation experiment within kit tubes. The experiment should however have been performed on subsamples taken from the experiment in 2.2/2.3 as the conditions in (closed?) kit tubes are very different from regular soil incubations. The authors attempt to link the results from both experiments, which in my opinion is not warranted as the experiments have been performed under different conditions.
    g. Section 2.4: for especially scenario E4, with an apparent excess of enzymes including DNase, I am surprised to see the authors report successful DNA purification. How have the authors achieved DNA purification in the presence of excess DNase? Idem for the scenarios with lower amount(s) of DNAse added?
2. Most results not significantly different from control experiments or have missing statistical analyses.
    a. The results of soil stability/SOM measurements indicate that none of the 'scenarios' are significantly different from the control experiment. The only significant difference the authors report concerns between-treatment results, which leaves me wondering about the relevance of the whole study.

b. The results shown in Figure 2 have been reported without statistical analyses on significant difference. Please include statistical analyses on significant difference between control and treatments. The figure's error bars of the control and the experimental treatments could suggest that differences between control and treatment scenarios are unlikely to be significant, leaving doubt about the experiment's relevance and study design.

c. Figure 3 is missing a control on DNA present in the added enzyme mixtures. Can the authors ensure that the DNA extracted and amplified is not derived from the enzyme preparations added? Especially scenario E4 might lead to addition of a lot of DNA.

d. Figure 3: In contrast to the above, DNase is added in the scenarios, which should then lead to degradation of DNA present in the samples. Can the authors therefore please clarify the puzzling details of this experiment?

e. Figure 3: Can the authors please provide (control) data on (expected) cell lysis from treatments, esp E4? This will enable untangling of results due to lysis and any EPS-biofilm effect on soil aggregation.

3. Discussion of results

a. The significant in-between-treatment results are given too much focus and attention, especially in the knowledge that none of the treatments were significantly different to controls. The majority of the conclusions drawn are not supported by the actual data provided.

b. Line 390 '… our results give a qualitative evidence for the influence of biofilms on aggregate stability…' This conclusion is not supported by the data provided.

c. Figure 4: this diagram can be omitted.

4. I am a bit puzzled by the section 'Data Availability', shouldn't these references be included in the References section? If this is provided according to the journal's instructions, then fine.

In conclusion, I have severe reservations regarding the study design and technical approach used in combination with the actual results (mostly not significantly different from controls). This in turn leads me to believe this manuscript cannot be improved substantially through major revision as a severe overhaul of study design and methodology is needed. My recommendation is therefore to reject the current manuscript for publication.

---

## Author Comment (AC1) · 3 May 2016

Enzymatic biofilm detachment causes a loss of aggregate stability in a sandy soil.

F. Büks1 and M. Kaupenjohann1

1 Chair of Soil Science, Department of Ecology, Technische Universität Berlin.

Correspondence to: F. Büks (frederick.bueks@tu-berlin.de)

Final response to Marc Redmile-Gordon (For the formatted document please open supplement)

Dear Mr Redmile-Gordon, first I would like to express my sincere thanks to you for reviewing, especially for your detailed and very helpful suggestions and your forbearance concerning grammatical errors.

[Figure]

Title

Line 1: Title should change, I suggest: 'Enzymatic biofilm digestion in soil aggregates facilitates the release of particulate organic matter (POM) by sonication'. We changed the title as suggested. Thank you very much.

General corrections

Lines 181, 200, 263, 264, 265, 266, 270, 272, 273, 280, 339, 340, 343, 371, 379, 426 and elsewhere: Renaming of SOC. As (1) C is the actual measure and (2) SOC involves DOC, which is rejected during POM extraction, POM and SOC are not suitable to term the C release from aggregates. Instead, "particulate organic carbon" (POC) will be used. This also includes organic molecules, already adsorbed on the HF after ultrasonic treatment. When describing the extracted material as a whole, POM will be used.

Lines 13, 14, 15, 26, 37, 39, 46, 49, 78, 79, 80, 83, 89, 92, 93, 96, 101, 105, 110, 112, 123, 138, 143, 144, 146, 147, 192, 210, 265, 280, 281, 283, 342, 343, 344, 406, 410, 413: Diverse suggestions to improve orthography, grammar, lucidity and scientific notification. All proposals are included. Thanks a lot.

Abstract

Line 24: delete 'which preserves aggregate structure'. "... which preserves aggregate structure, ..." was removed, as additional influence on binding mechanisms such as surface charge of POM cannot be ruled out.

Line 30: This is overly confident and not quite accurate. Is it not true that enzymatic digestion of EPS polymers may have increased the abundance of EPS fragments released upon sonication? Therefore, remove 'our results confirm, that EPS stabilises soil aggregates predominantly by a strong intra-aggregate fixation, and enzymatic biofilm digestion caused a shift of occluded particulate organic matter (POM) to more fragile binding patterns' and replace with 'our results suggest that EPS stabilises intra-aggregate particulate organic matter (POM) within soil aggregates'. The samples have a Cmic of 0.352 mg g-1 dry soil aggregates and a Corg of 8.7 mg g-1 dry soil. total POC release amounts to 1.8 mg g-1 dry soil for E0 and 1.98 mg g-1 dry soil for E4 – the difference (0.18 mg g-1 dry soil) is half the Cmic. Therefore it is (mathematically) possible, that the whole difference in POC release is caused by release of biofilm fragments. The real share is unknown, but the small share of released bacterial DNA as well as visibly increased dark POM release in E4 after sonication reinforce additional non-biofilm POM release. However, we choose the more careful statement as you suggested "our results suggest that EPS stabilises intra-aggregate particulate organic matter (POM) within soil aggregates" and will revisit this in the discussion part.

Introduction

Lines 61-63: awkward sentence, please rephrase. Done: "In addition, carbonates and phosphates as well as microbial precipitates force up aggregation."

Line 82: replace 'biofilm forming species and habitats:' with 'community composition and environmental cues:' Sounds much better. Thank you and done.

Line 108: Unsubstantiated statement which leaves the reader wondering 'why'. I suspect the authors are drawing on the rationale presented Redmile-Gordon et al. (2014) and suggest this is expanded upon for clarity and to help build justification. Suggest the authors replace 'That is mainly due to methodological reasons' with 'This is mainly due to methodological reasons. For example, Tang et al. (2011) found no link between bacterial EPS extracted using sulphuric acid and aggregate stability. Redmile Gordon et al (2014) subsequently found in a comparison study that the techniques previously used to measure extracellular polysaccharide in soil co-extracted large quantities of 'random' soil organic matter which confounded estimates of EPS production." I will add "Though Tang et al. (2011) showed a significant contribution of bacterial growth on aggregate stability, the observations could not definitely be attributed to soil microbial exopolysaccharide production. Redmile-Gordon et al. (2014) subsequently found

that the techniques previously used to measure extracellular polysaccharide in soil co-extracted large quantities of 'random' soil organic matter which confounded estimates of EPS production."

Material and Methods

Lines 141-142: This is not a method to estimate soil microbial biomass, this is respiration, correct accordingly. "To estimate the soil microbial biomass" refers to the whole paragraph. For clarification, the paragraph will be reshaped to "To estimate the soil microbial biomass, first 8 x 10 g of soil aggregates have been adjusted to 70 vol% soil water content and incubated for 70 hours at 20°C in the dark to attain basal respiration. Then, based on DIN EN ISO 14240-2 ..."

Lines 163-172: This section takes some time to understand. Insert "sufficient enzymes were provided to digest the EPS content expected in five scenarios (E0 to E4)" Line 165: "each" added before "with highest need of enzymatic units for the total biofilm detachment". Line: 172: "five scenarios were design" is replaced with "sufficient enzymes were provided to digest the EPS content expected in five scenarios:"

Lines 181/193: ... e.g. Cerli et al 2012 do not claim this method quantifies aggregate stability Cerli et al. (2012) was replaced by Golchin et al. (1994) as prime reference. Cerli et al. (2012) will appear in the discussion about light fraction release as indicator of aggregate stability.

Line 190: Why for 30 min? To allow NaPT diffusion? "... to allow SPT diffusion into the aggregates" will be added.

Line 195: 50 J ml-1 given over what time period? Time periods depend on the weight of sample+SPT solution and fluctuate around 1 min 15 sec.

Line 217: What volume of wash was used as an equivalent for the mass of soil stipulated in the FastDNATM spin kit soil manual? (Can it really be used to extract DNA from a dilute wash and compare with soil?) FastDNA^TM SPIN KIT (used for for liquid
samples of 200 $\mu$l and pure cultures) and FastDNA TM SPIN Kit for Soil (normally used with "Up to 500 mg of soil sample" ansd for complicated samples) only differ (1) in the first buffer, (2) the point of time for the application of protein precipitation solution (PPS) and (3) in the last incubation procedure (incubation in DES solution for 5 minutes in a heat block at 55°C after addition of SEWS-M instead of incubation at room temperature before addition of DES). Both methods are very similar. As we did a qualitative comparison of DNA release, variance of DNA release between methods is of minor importance.

Results

Line 268: Move 'data are shown as mean values and standard deviations of five parallels' to figure caption Done.

Lines 274-279: Incorrect (and potentially misleading) presentation of results. Suggest as replacement: "there was no increase or decrease relative to the control, however, there was a trend for increased POM release with increasing enzyme addition, and the difference between the lowest enzyme addition and the highest was statistically significant as indicated by the Tukey test. This trend was only broken by the control treatment (given no enzymes)" // Unnecessary and confusing statement, we can see the standard deviation and Tukey test results on the figure, better to remove the statement. // Potentially misleading statement, yes, E2 and E3 have no difference compared to the control, but neither do E1 or E4. Thank you very much for the proposal. We decided to desist from a specific significance level in the revision of this paper. A p-value of 0.05 is a convention underpinned only by practical but not scientific reason. E.g. visible differences are leveled by using it: E0, E2 and E3 appear to have similar mean values and variance, whereas E1 (p=0.6) and E4 (p=0.15) show visible differences to the control at 50 J/ml. Whereas E1 is not explained by the model and have to be discussed, E4 matches the forecast and is underpinned by the increase in bacterial cell release. That has to be carefully discussed. "There was a trend for increased POC release with increasing enzyme addition, and this trend was only broken by the control treatment

(E0, given no enzymes)."

Lines 284-288: There has been no physical transfer of organic matter between these analytical pools. A reduced aggregate stability may have for example, or increased release of biofilm fragments retained on the 1.5 $\mu$ glass filter, but this is a matter for discussion. It might be more useful to say here that it is reassuring that the SOC remaining in the sediment reflects what would be expected given the quantities extracted at 50 J... but of course it would (because you present relative fractions in preference to absolute concentrations). I am struggling to find a reason to retain this section. I think it better to delete lines. Our intention was to express that nearly the whole net POC differences E1-E0 and E4-E0 are related to variations in the HF, but not in fLF, oLF(100) and oLF(150). That will be included in lines 274-297.

Line 289: and Figure 2 These results have already been presented, it is not clear exactly what compounded estimate of error is being given, and besides, data were already presented in figure 1. Remove Figure 2. Line 290: This has already been presented, that one can add the non-significant results to the significant, and finds the same thing is nothing surprising or worthy of comment. Delete. Line 291: Clumsy sentence and repetition: delete first sentence. And lines 291-293: Released POM data may be evidence of this, and may not be - this is a matter for the discussion. Delete these lines. Lines 289-293 will be deleted. Fig. 2 will be changed to mg POC /g dry soil and shortly described.

Lines 293-296: Delete section starting "The lower aggregate stability is indicated by a steeper gradient and on average in an...". Replace with "The addition of the highest enzyme concentration (E4) caused the release of about 40% more POM by mild sonication (50J ml-1) than occurred with the addition of the lowest concentration (E1). This was statistically significant at (p <0.05)." end of section. Thank you very much. "At 50 J ml-1 ultrasonic treatment results in an additional POC release of about 10% more POC compared to the control, whereas POC release is reduced by -18% in E1. The addition of the highest enzyme concentration (E4) caused the release of about 1/3

more POM by mild sonication (50 J ml-1) than occurred with the addition of the lowest concentration (E1) (p=0.003).

Line 302: In contrast here I think the relative increase in DNA release is a little understated. Yes it is useful to also give it as a percentage of total DNA extracted from the soil as you have done (Figure 3 - now rename to Figure 2), but perhaps in line 302 replace text "it is increased by about 3.5% to a value of 5.5% in the E4 scenario in comparison to the control" with 'While there was no difference in DNA concentrations suspended in the wash of control and low enzyme additions, treatment E4 caused an increase to more than double the DNA content of either E0 or E1." Thank you. Replaced by: "While there was no difference in relative DNA release in the wash of control and low enzyme additions, treatment E4 caused an increase to more than double the DNA content of either E0 or E1, which amounts to 5.6% of total DNA".

Discussion

Lines 324-335: First paragraph disorganised: it is an unpleasant jump to the model in the first sentence. Build up to it. It would be smoother if begin with the main result result, followed by your description of enzyme transport into the unsaturated pore space and discussion of others work E.g. "We found that increasing the quantity of enzymes applied to aggregates led to increased release of POM when aggregates were sonicated. Then describe the pore system (currently lines 325 326), then give your model of explanation "we present a model to explain the observed findings ..." Thank you. First paragraph was replaced by: "We found that increasing the quantity of enzymes applied to aggregates led to increased release of POC when aggregates were sonicated. This detachment is explained by the transport of $\alpha$-glucosidase, $\beta$-galactosidase, DNAse and lipase into the unsaturated pore space. Consequently enzymes diffuse into the biofilm matrix, where structural components like polysaccharides, eDNA and lipids are digested as approved for diverse enzymes and enzyme targets in ecological and medical studies (Böckelmann et al., 2003; Walker et al., 2007). We utilize a simple spacial model to explain the observed findings: The biofilm bridges gaps between primary

particles, connects them and builds a restructured pore system inside the aggregate (Fig. 4). As macromolecular biofilm components yield EPS as a viscoelastic structure (Sutherland, 2001), their digestion causes a loss in EPS viscosity and thereby should reduce aggregate stability. The effect is expected to grow with increasing enzyme activity until the whole EPS matrix is dispersed."

Lines 336-337: Delete the discussion of what is not being discussed. Done.

Line 345: 'de facto' is way too strong and encourages the reader think of examples to disprove this over-confident statement. E.g. it could have been caused by cell lysis. Delete 'de facto'. Done.

Line 352: This is not the only possible explanation and further discussion with relevant literature is required. Might some of the C released from occluded POM and/or biofilm not have been detected in the filtered light fraction? – e.g. may have been present as smaller particulates or DOC? Also, DNA/cells/POM may not have been released without sonication. Include this. Current literature has more to offer. Add "Furthermore, we pre-incubated soils given 0.2 mM NH4NO3, and added further NH4NO3 with the enzyme application. Redmile-Gordon et al (2015) proposed that low C/N ratios of substrates available to soil microorganisms reduces cell specific EPS production rates, and may trigger microbial consumption of EPS to acquire C for cell-growth. The observations leading to this proposed dynamic were also found by addition of NH4NO3. In the present study, NH4NO3 was applied with all treatments including the control (which also received no C from enzyme provision). The resulting lowest C/N ratio in the control soils may itself have decreased the EPS, contributing to the higher than expected release of POM from the control soil with sonication at 50 J mL-1, and the break in the trend for increasing POM release with increasing enzyme addition. We now write: "Decreased POC release in E1 could be explained by pre-incubation of soil aggregates given 0.2 mM NH4NO3 and further addition of NH4NO3 with enzyme application. Redmile-Gordon et al. (2015) proposed that low C/N ratios of substrates available to soil microorganisms reduces cell specific EPS production rates, and may

trigger microbial consumption of EPS to acquire C for cell-growth. The observations leading to this proposed dynamic were also found by addition of NH4NO3. In the present study, NH4NO3 was applied with all treatments including the control (which also received no C from enzyme provision). The resulting lowest C/N ratio in the control soils may itself have decreased the EPS, contributing to the higher than expected release of POM from the control soil with sonication at 50 J mL-1, and the break in the trend for increasing POM release with increasing enzyme addition." Further "Probably high enzyme concentrations dissolve biofilm structures that remain part of the coarse POM at low enzyme treatment, which results in underestimation of E4 POC release." was added in this paragraph.

Lines 350-352, 390: Discussion about link between biofilm digestion and aggregate stability. Sentence in lines 350-352 "The incomplete biofilm digestion suggests, that the influence of biofilms on aggregate stability is larger than demonstrated in scenario E4." shifted to a later part of discussion. Previous reference to aggregate stability is replaced by biofilm digestion/POC release context, except in the spacial model.

Line 353: Replace "The incomplete ... ambiguously" sentence with "Nonetheless, biofilm detachment caused by E4 is still likely to be incomplete." And continue with "Slow enzyme diffusion..." Line 348-350 is replaced by "Hence, biofilm detachment caused by E4 is still likely to be incomplete." Continued with "Slow enzyme diffusion...".

Lines 352, 356-367: This paragraph contains some useful information that should be retained for comparison of enzyme quantities added. However, the explanation drawing on enzyme activities in natural soils is not clear and needs re-thinking and re-writing. Actually, it seems the argument is flawed. You only observed effects when you increased enzyme activities well above 'natural' levels so on the contrary seems to support the hypothesis that diffusion factors ARE limiting (e.g. sorption to active surfaces). Suggest you cite the excellent review by (Burns et al., 2013) (see section 3.3; page 220). "Based on our calculations enzyme concentrations of mix E1 should be sufficient for total biofilm digestion within time of application (1h) – as far as there are

no other factors reducing enzyme efficiency. As surveys of natural soils show enzyme concentrations up to mix E3 [Cooper and Morgan, 1981; Eivazi and Tabatabai, 1988; Acosta-Martinez and Tabatabai, 2000], such factors might be reasonably assumed. This is underpinned by our results, that show the only increase in POC release in scenario E4 attended by only an incomplete cell release. After addition to the soil sample, enzymes must enter the EPS matrix by diffusion. Therefore it is assumed that parts of the enzymes probably do not reach the biofilm due to inhibited diffusion. Beside diffusion, sorption and decomposition could play a major role in reducing enzyme efficiency. Whereas turn-over rates of soil enzymes are not yet assessed, extended stabilization of active enzymes over time on soil mineral and organic surfaces is reported (Burns et al., 2013). This mechanism could explain immobilization of enzymes off the biofilm and high measured soil enzyme concentrations from literature in face of still existing biofilms. Due to this boundary conditions, quantification of the relation of enzyme concentration and POC release was not possible in this work, although there is a tendency for enhanced POC release." This information will be included in lines 356-367.

Lines 368-370: It does not reinforce this, and if it does it conflicts with your model. If your model is correct it would only be found'after disruption of aggregates to release the oLF (as you observed at 50 J ml-1; congruent with your model). It could also have been lost as soluble C, as mentioned above in reference to line 352 above. Delete 368 – 370. Correct. I referred to a POM occlusion only mediated by EPS, but in the model it seems very implausible to assume occluded POM, that is not bound by physico-chemical interactions. Deleted.

Lines 378-383: Not statistically significant therefore remove this speculation. Statistically it is built on observations that can be reasonably expected by chance. Done.

Line 384 replace 'cumulation of LF carbon release overall energy level clarifies the alteration of soil aggregate stability' with 'The trend for increased of LF carbon release over increasing enzyme additions demonstrates an alteration of soil aggregate stability'. Thank you. Done.

Line 385 – results repetition. And lines 386-389 Careful, you are discussing SOC (POM) release and aggregate stability as if you measured both independently, and focus drifts. I recommend you instead discuss the connection you propose (POM release being due to digestion of EPS which seems to prevent POM release by sonication alone up to 150 J ml-1 – and after more effectively separated from soil minerals by 50J sonication). See "New line of argument" at the end of this document.

Line 395: Good point re enzyme metabolism, although 1 hour is not a lot of time for it, it would be useful to include a reference for rapid metabolism of enzymes/proteins. Add that the large additions of enzyme-C could be used as a C-source for microbial growth which is known to stabilise soil aggregates, e.g. (Watts et al., 2005). This is why total enzyme-C added should be included in your manuscript (suggest this is added to Table 3). "The applied enzymes have no relevant mass input to extractable POM. Even in case of complete adsorption to POM in only one fraction, highest enzyme concentration (E4) would result in additional 13.5 $\mu$g enzyme /g dry soil being <0.4% of the smallest extracted POM fraction. Although enzyme concentration has no influence on extracted POC, addition of enzyme-C could be used as microbial metabolic C-source which is known to lead to soil aggregate stabilization (Watts et al., 2005; Tang et al., 2011). Soil turn-over rates of enzymes are not assessed (Burns et al., 2013). Fast metabolization of enzymes within 1 hour would hinder quantification of the relation of biofilm digestion and POC release by influencing aggregate stability during the experiment." This content will be connected to point (Line 352).

Lines 407, 408: better if you delete 'a 9000 fold of the E1 enzyme activity calculated from actual soil biomass to remove approximately // suggest replace '5.5% of the biofilm and no increase in FLF release, the pooled influence of the disregarded boundary conditions on enzymatic detachment efficiency is large' with '5.5% biofilm removal indicated by DNA measurements coupled with no increase in fLF release, may suggest that the pooled influence of the disregarded boundary conditions on enzymatic detachment efficiency is large'. As the role of fLF C is discussed regarding lines 368-370, the

paragraph is replaced with: "Most of these restrictions are owed to the high complexity of the soil ecosystem. Enzymes were applied in concentrations four orders of magnitude higher than calculated from actual Cmic and even 1-2 orders of magnitude higher than values from literature. Considering maximum 5.5% biofilm removal indicated by DNA measurements may suggest that the pooled influence of the disregarded boundary conditions on enzymatic detachment efficiency is large." Orders of magnitude are still noted to illustrate the probable range of influence of the disregarded boundary conditions.

Lines 410-413: delete 'nonetheless' // replace 'Loss of aggregate stability' with 'Release of entrapped POM' // replace 'stabilisation' with 'stabilising' // Citation needed: suggest after 'stabilising agent of soil aggregates' to insert 'as discussed in a comprehensive review by Or et al. (2007)'. // Subsequent sentence, why limit to just natural ones? I suggest you replace 'Aggregate stability is influenced by the digestion of EPS components. Adapting this relation to natural soil ecosystems,'" with 'The apparent loss of aggregate stability caused by the digestion of EPS components in the present study suggests biofilm relevance in soil ecosystems.' And finish the discussion there. Paragraph replaced with "These results give insight in fundamental processes underlying aggregate stability. Release of occluded POM coupled with increased bacterial DNA release after treatment with high enzyme concentrations underpin the assumption that biofilm is a stabilising agent of soil aggregates as discussed in a review by Or et al. (2007). The apparent loss of aggregate stability caused by the digestion of EPS components in the present study suggests biofilm relevance in soil ecosystems e.g. in terms of soil-aggregate related functions like soil water dynamics, mechanical stability as well as rootability."

Conclusion

Lines 414-417, 419-420, 422-423, 425, 425-427, 427, 431: Move this final part to the start of conclusions: "Our results suggest a change of biofilm composition due to a shift ..." // Already discussed, is weak, better to delete. // delete "and thereby enhances

aggregate stability". Already discussed and now superseded by your two important sentences above this (first one suggested to be taken from discussion, lines 414 – 417). // Delete 'fLF' (these abstract technical distinctions are not appropriate for this statement). Continue with the condition i.e. "not to an increase in fLF release without physical disruption of aggregates by sonication." // replace SOC with POM (should already be defined) 427 delete the sentence starting "The bacterial DNA..." as discussed already; this does not withstand logical critique. // 'microbial communities' already are for various reasons, I think you mean the biofilm or EPS, EPS being relevant even when no biofilm can be observed . . . suggest you replace 'communities' with 'EPS dynamics'. New conclusion: "It was shown that EPS is a factor of aggregate stability. Our experimental results suggest that extracellular polymeric substance (EPS) contributes to occlusion and attachment of particulate organic matter (POM) in soil aggregates. The application of a highly concentrated mix of $\alpha$-glucosidase, $\beta$-galactosidase, DNAse and lipase is related to a detachment of POM from a stable to a more fragile binding structure, but not to an increase in POM release without physical disruption of aggregates by sonication. The pattern of measured POC release and additional bacterial DNA release points to an intra-aggregate fixation of POM by enzyme targets. A loss of EPS integrity could therefore cause a detachment of soil organic matter, not only in the laboratory but also in natural soil ecosystems. Our results further suggest that a change of biofilm composition probably due to a shift in microbial population structure may alter soil aggregate stability. On macro-scale this could affect soil compactibility, erodibility, water transport, retention and aeration regime, rooting depth and the occlusion of soil organic carbon. This, in conclusion, invites to behold soil EPS dynamics as a factor of sustainable land use."

Figures and Tables

Figure 4: edit caption – you are not showing 'biofilm structure' – this is 'aggregate structure' replace accordingly. Caption changed to "Proposed model of aggregate structure: ..."

Table 3: Add quantity of enzyme-C added to enable judgement of substrate utilisation by soil microbial biomass. Quantities added.

Table 3: column E0: should the q value not be zero? Otherwise why are the enzyme activities different from column E1? Yes. Thanks.

Furthermore ... … there are also some points I have to answer back.

Line 38: insert 'and' before 'is an integral' That doesn't fit in this place.

Line 56: delete '.' There is an end of sentence and the references are related to the whole paragraph.

Line 173: use large 'C' for carbon c in ccell means "concentration"

Line 409: Insert sentence: 'Conversely, or in addition to the above, complete biofilm removal may have been achieved, however as the model (figure 4 – now figure 3) proposes, POM would not be released until the retaining aggregates were disrupted by disruptive physical forces such as those caused by sonication.' (Kaiser and Berhe, 2014) As only 5.5% of the bacterial DNA are removed after enzymatic treatment, it seems implausible to expect complete biofilm detachment. Further, point (Lines 368-370).

New line of argument Line of argument will be restructured in the following way (e.g. to avoid repetitions): Discussion of POM release (increase in E4, decrease in E1, tendency, p-values but no significance level) – discussion of bacterial DNA release – discussion of of the relation of both (EPS as enzyme target) – discussion of the explanatory power of (small) POM release and of its usability for aggregate stability measurement in similar soil samples – "A more quantitative analysis of the relation of enzymatic EPS detachment and POM release would require more replicate samples and probably inclusion of soils from different land use. However, this was beyond the scope of the present study."

Best regards, Frederick Büks

Please also note the supplement to this comment:
http://www.soil-discuss.net/soil-2015-87/soil-2015-87-AC1-supplement.pdf

---

## Author Comment (AC2) · 3 May 2016

Enzymatic biofilm detachment causes a loss of aggregate stability in a sandy soil.

F. Büks1 and M. Kaupenjohann1

1 Chair of Soil Science, Department of Ecology, Technische Universität Berlin.

Correspondence to: F. Büks (frederick.bueks@tu-berlin.de)

Final response to Referee2 (For the formatted document please see supplement)

Dear Referee2. Thank you very much for reviewing. In the following I will try to answer your important comments and to clear the objections.

1.a. Section 2.2: confusingly written maths section Pooling of equations to a single

one is a space saving way to show these manifold steps of converting concentrations of biofilm components to the final value of needed enzyme units. I will place each single step in the supplements. The paragraph beginning in line 163 will be revised to clarify the following: Literature show a wide range of enzyme-target concentrations in different soils. As we do not know target concentrations of our soil (due to a lack of extraction methods), we considered the largest published concentrations to find existing effects. Further as target molar masses vary as well, here we choose the smallest mass. Both conduce to a "worst-case" point of view with maximum enzyme targets.

1.b. Section 2.2: poor justification of numbers used: Eg the supposed soil bulk density number seems odd, as this can be measured for field core samples and be recreated to field soil density. Otherwise explain the assumption for this particular experiment as normal dried and sieved soil without repacking does not get to this density. Different samplings during the field experiment showed soil bulk densities of 1.4 g/cm3. These values are normal for a sandy silt (Su3) [Chaudhari et al., 2013], that is used in this experiment. For scenario E1 soil bulk density is irrelevant because ccell and therefore target maxima were estimated from cmic. For scenario E2 and the following we measured a minor soil bulk density in a sample of soil aggregates (∼1.15 g/cm3). On the other hand, biofilm populations are mentioned to be mainly located in soil aggregates [Nunan et al., 2003]. Therefore – following our "worst-case"-approach – we used the bulk density of the original soil to estimate maximum target values.

1.c. Section 2.2/2.3: poor justification of numbers used: The 'scenarios' have been explained (though could be improved in clarity) but do not actually contain any information regarding the technical set up. How much enzyme activity units were applied? Enzyme units are listed in table 3. What was the level of purity of the enzyme preparations? Enzyme purity is guaranteed by the producer (data sheets of product numbers Sigma-Aldrich: G0660, G5635, L0382 and D5025). How where the enzymes added? Was there mixing involved? Enzymes were added as described in section 2.4. Enzyme solutions were vortexed and than added to aggregate samples as described in section

2.4. There is a severe lack of information, especially as the whole manuscript depends on contact of these enzymes with EPS materials. How have the authors assured that these enzymes have reached the materials processed further? Contact of enzymes and EPS on the micro-scale were not demonstrated directly. Contact of enzymes to EPS can be assumed as the whole enzyme solution was absorbed by the soil aggregates. Fine pores (already filled with ARW from pre-incubation) in contrast need to be supplied by diffusion, that is probably inhibited. However, enzymes are able to diffuse into the EPS within 1 hour, as described by Böckelmann et al. (2003). Thus, observed effects are not quantitative, but qualitative. We tend to express the more cautious position "Enzymatic treatment causes an increased release of POM after sonication" to include uncertainties about enzyme contact to targets.

1.d. Section 2.2/2.3: the E4 scenario seems to suggest a large excess of enzymes was applied. How have the authors ensured that such a large excess is not damaging to resident live microbial cells? E.g. a large excess of lipase may affect the membrane integrity of cells. This may in turn impact on DNA quantification without actually directly affecting soil aggregate stability. Cell membranes are built of phospholipids. We used purified lipase from porcine pancreas. Lipases are cutting fatty acids off e.g. glycerol, but are unable to cut fatty acids from phospholipids (as phospholipases do).

1.e. Section 2.3: information/studies on basal respiration at 30C/37C, the temperature of the actual experiments performed, are missing. Respiration data are collected at $20°C$ in another experiment using the same soil, where basal respiration was reached after 2 days. Therefore we concluded 3 days as sufficient to reach basal respiration at even higher temperatures.

1.f. Section 2.4: this experiment was performed on a separate soil incubation experiment within kit tubes. The experiment should however have been performed on subsamples taken from the experiment in 2.2/2.3 as the conditions in (closed?) kit tubes are very different from regular soil incubations. The authors attempt to link the results from both experiments, which in my opinion is not warranted as the experiments have

been performed under different conditions. A direct subsampling from the aggregate stability experiment to perform the DNA experiment was rejected due to its destructive capability regarding aggregates. In turn, temperature, substrate, pH and water content of the tube experiment were similar to the incubation of samples for the measurement of aggregate stability. Further differences were disregarded. As part of our hypothesis, the link between both increase in POM release and bacterial cell release can explained causally.

1.g. Section 2.4: for especially scenario E4, with an apparent excess of enzymes including DNase, I am surprised to see the authors report successful DNA purification. How have the authors achieved DNA purification in the presence of excess DNase? Idem for the scenarios with lower amount(s) of DNAse added? During incubation DNAse only digest free DNA but not DNA within bacterial cells of the biofilm. Later the pooled wash solution contains most of the DNAse. After centrifugation, this solution was discarded, whereas the bacterial pallet was resuspended in 200 $\mu$l ARW. At this stage, bacterial cells are still intact and immune to DNAse, whereas DNAse is diluted and hindered by high buffer ion concentrations. After mechanical cell lysis PPS (Protein Precipitation Solution) was added leading to e.g. precipitate DNAse. All steps of DNA extraction were conducted on ice to strongly reduce enzyme activity.

2.a. The results of soil stability/SOM measurements indicate that none of the 'scenarios' are significantly different from the control experiment. The only significant difference the authors report concerns between treatment results, which leaves me wondering about the relevance of the whole study. Even if there is no significant difference in aggregate stability, unfulfilled expectations (as e.g. a dramatic loss in aggregate stability after enzymatic treatment) do not minder the relevance of a study. In addition – without any attempt to prettify our results – p-values <0.05 as the limit for significance is a convention. From my point of view there is a tendency of increasing POM release (p=0.1, 5 parallels) in E4, and a tendency to decrease (p=0.06, 5 parallels) in E1 compared to the control. The first one fits to our model, the second one does not. Both

tendencies are visible and have to be explained under the restriction of being small.

2.b. The results shown in Figure 2 have been reported without statistical analyses on significant difference. Please include statistical analyses on significant difference between control and treatments. The figure's error bars of the control and the experimental treatments could suggest that differences between control and treatment scenarios are unlikely to be significant, leaving doubt about the experiment's relevance and study design. Good idea. I will do this. Thereby, y-axis of figure 2 will be converted to mg POC /g dry soil.

2.c. Figure 3 is missing a control on DNA present in the added enzyme mixtures. Can the authors ensure that the DNA extracted and amplified is not derived from the enzyme preparations added? Especially scenario E4 might lead to addition of a lot of DNA. We do not have this data. ARW for stock solutions and dilutions include ultrapure water and were autoclaved. Remaining free DNA strand amount is assumed to be far below soil DNA concentration and most probably digested by DNAse in stock and dilutet solutions. Further possible DNA additions were similar between variants and related to blind values.

2.d. Figure 3: In contrast to the above, DNase is added in the scenarios, which should then lead to degradation of DNA present in the samples. Can the authors therefore please clarify the puzzling details of this experiment? Until mechanical cell lysis, extracellular DNA including eDNA from EPS is digested by DNAse. As mentioned in 2.c. small amounts of additional DNA are supposed to be irrelevant and the bulk of free DNA is rejected by washing. E4 shows the highest DNA release, although undigested biofilm in low enzyme treatments could increase DNA-concentration via centrifugation to the pallet. That could probably point to an underestimated additional DNA release in E4. It underlines the only qualitative approach of this experiment.

2.e. Figure 3: Can the authors please provide (control) data on (expected) cell lysis from treatments, esp E4? This will enable untangling of results due to lysis and any

EPS - biofilm effect on soil aggregation. DOC release from bacterial cells due to enzymatic treatment and ultrasonication was not quantified. This DOC is most probably removed by repeated washing during density fractioning. Measuring the distribution of remaining bacterial DOC among fLF, oLF and HF is impossible by method. However, visibly increasing POM release in E4 (see Line 30, Final response to Marc Redmile-Gordon) points to a negligible effect of bacterial DOC sorption on measured C.

3.a. The significant in – between - treatment results are given too much focus and attention, especially in the knowledge that none of the treatments were significantly different to controls. The majority of the conclusions drawn are not supported by the actual data provided. and 3.b. Line 390 '... our results give a qualitative evidence for the influence of biofilms on aggregate stability...' This conclusion is not supported by the data provided. See point 2.a. We propose careful line of argument including a statement of insignificance and a discussion of tendencies in face of p-values nearby p=0.05.

3.b. Figure 4: this diagram can be omitted. As figure 4 illustrates the model, we prefer to retain it.

Best regards, Frederick Büks

Please also note the supplement to this comment:
http://www.soil-discuss.net/soil-2015-87/soil-2015-87-AC2-supplement.pdf

**Supplement:**

**Enzymatic biofilm detachment causes a loss of aggregate stability in a sandy soil.**

**F. Büks[1] and M. Kaupenjohann[1]**

[1] Chair of Soil Science, Department of Ecology, Technische Universität Berlin.

*Correspondence to:* F. Büks (frederick.bueks@tu-berlin.de)

**Final response to Marc Redmile-Gordon**

*Dear Mr Redmile-Gordon,*
*first I would like to express my sincere thanks to you for reviewing, especially for your detailed and very helpful suggestions and your forbearance concerning grammatical errors.*

**Title**

**Line 1:** *Title should change, I suggest: 'Enzymatic biofilm digestion in soil aggregates facilitates the release of particulate organic matter (POM) by sonication'.*
We changed the title as suggested. Thank you very much.

**General corrections**

**Lines 181, 200, 263, 264, 265, 266, 270, 272, 273, 280, 339, 340, 343, 371, 379, 426 and elsewhere:** *Renaming of SOC.*
As (1) C is the actual measure and (2) SOC involves DOC, which is rejected during POM extraction, POM and SOC are not suitable to term the C release from aggregates. Instead, "particulate organic carbon" (POC) will be used. This also includes organic molecules, already adsorbed on the HF after ultrasonic treatment. When describing the extracted material as a whole, POM will be used.

**Lines 13, 14, 15, 26, 37, 39, 46, 49, 78, 79, 80, 83, 89, 92, 93, 96, 101, 105, 110, 112, 123, 138, 143, 144, 146, 147, 192, 210, 265, 280, 281, 283, 342, 343, 344, 406, 410, 413:** *Diverse suggestions to improve orthography, grammar, lucidity and scientific notification.*
All proposals are included. Thanks a lot.

**Abstract**

**Line 24:** *delete 'which preserves aggregate structure'.*
"… which preserves aggregate structure, …" was removed, as additional influence on binding mechanisms such as surface charge of POM cannot be ruled out.

**Line 30:** *This is overly confident and not quite accurate. Is it not true that enzymatic digestion of EPS polymers may have increased the abundance of EPS fragments released upon sonication? Therefore,*

*remove 'our results confirm, that EPS stabilises soil aggregates predominantly by a strong intra-aggregate fixation, and enzymatic biofilm digestion caused a shift of occluded particulate organic matter (POM) to more fragile binding patterns' and replace with 'our results suggest that EPS stabilises intra-aggregate particulate organic matter (POM) within soil aggregates'.*

The samples have a $C_{mic}$ of 0.352 mg $g^{-1}$ dry soil aggregates and a $C_{org}$ of 8.7 mg $g^{-1}$ dry soil. total POC release amounts to 1.8 mg $g^{-1}$ dry soil for E0 and 1.98 mg $g^{-1}$ dry soil for E4 – the difference (0.18 mg $g^{-1}$ dry soil) is half the $C_{mic}$. Therefore it is (mathematically) possible, that the whole difference in POC release is caused by release of biofilm fragments. The real share is unknown, but the small share of released bacterial DNA as well as visibly increased dark POM release in E4 after sonication reinforce additional non-biofilm POM release. However, we choose the more careful statement as you suggested "our results suggest that EPS stabilises intra-aggregate particulate organic matter (POM) within soil aggregates" and will revisit this in the discussion part.

**Introduction**

**Lines 61-63:** *awkward sentence, please rephrase.*
Done: "In addition, carbonates and phosphates as well as microbial precipitates force up aggregation."

**Line 82:** *replace 'biofilm forming species and habitats:' with 'community composition and environmental cues:'*
Sounds much better. Thank you and done.

**Line 108:** *Unsubstantiated statement which leaves the reader wondering 'why'. I suspect the authors are drawing on the rationale presented Redmile-Gordon et al. (2014) and suggest this is expanded upon for clarity and to help build justification. Suggest the authors replace 'That is mainly due to methodological reasons' with 'This is mainly due to methodological reasons. For example, Tang et al. (2011) found no link between bacterial EPS extracted using sulphuric acid and aggregate stability. Redmile Gordon et al (2014) subsequently found in a comparison study that the techniques previously used to measure extracellular polysaccharide in soil co-extracted large quantities of 'random' soil organic matter which confounded estimates of EPS production."*
I will add "Though Tang et al. (2011) showed a significant contribution of bacterial growth on aggregate stability, the observations could not definitely be attributed to soil microbial exopolysaccharide production. Redmile-Gordon et al. (2014) subsequently found that the techniques previously used to measure extracellular polysaccharide in soil co-extracted large quantities of 'random' soil organic matter which confounded estimates of EPS production."

**Material and Methods**

**Lines 141-142:** *This is not a method to estimate soil microbial biomass, this is respiration, correct accordingly.*
"To estimate the soil microbial biomass" refers to the whole paragraph. For clarification, the paragraph will be reshaped to "To estimate the soil microbial biomass, underline{first} 8 x 10 g of soil aggregates have been adjusted to 70 vol% soil water content and incubated for 70 hours at 20°C in the dark to attain basal respiration. Then, based on DIN EN ISO 14240-2 ..."

**Lines 163-172:** This section takes some time to understand. Insert "sufficient enzymes were provided to digest the EPS content expected in five scenarios (E0 to E4)"
Line 165: "each" added before "with highest need of enzymatic units for the total biofilm detachment".

Line: 172: "five scenarios were design" is replaced with "sufficient enzymes were provided to digest the EPS content expected in five scenarios:"

**Lines 181/193:** *… e.g. Cerli et al 2012 do not claim this method quantifies aggregate stability*
Cerli et al. (2012) was replaced by Golchin et al. (1994) as prime reference. Cerli et al. (2012) will appear in the discussion about light fraction release as indicator of aggregate stability.

**Line 190:** *Why for 30 min? To allow NaPT diffusion?*
"... to allow SPT diffusion into the aggregates" will be added.

**Line 195:** *50 J ml$^{-1}$ given over what time period?*
Time periods depend on the weight of sample+SPT solution and fluctuate around 1 min 15 sec.

**Line 217:** *What volume of wash was used as an equivalent for the mass of soil stipulated in the FastDNATM spin kit soil manual? (Can it really be used to extract DNA from a dilute wash and compare with soil?)*
FastDNA™ SPIN KIT (used for for liquid samples of 200 µl and pure cultures) and FastDNA TM SPIN Kit for Soil (normally used with "Up to 500 mg of soil sample" ansd for complicated samples) only differ
(1) in the first buffer,
(2) the point of time for the application of protein precipitation solution (PPS) and
(3) in the last incubation procedure (incubation in DES solution for 5 minutes in a heat block at 55°C after addition of SEWS-M instead of incubation at room temperature before addition of DES). Both methods are very similar. As we did a qualitative comparison of DNA release, variance of DNA release between methods is of minor importance.

**Results**

**Line 268:** *Move 'data are shown as mean values and standard deviations of five parallels' to figure caption*
Done.

**Lines 274-279:** *Incorrect (and potentially misleading) presentation of results. Suggest as replacement: "there was no increase or decrease relative to the control, however, there was a trend for increased POM release with increasing enzyme addition, and the difference between the lowest enzyme addition and the highest was statistically significant as indicated by the Tukey test. This trend was only broken by the control treatment (given no enzymes)" // Unnecessary and confusing statement, we can see the standard deviation and Tukey test results on the figure, better to remove the statement. // Potentially misleading statement, yes, E2 and E3 have no difference compared to the control, but neither do E1 or E4.*
Thank you very much for the proposal. We decided to desist from a specific significance level in the revision of this paper. A p-value of 0.05 is a convention underpinned only by practical but not scientific reason. E.g. visible differences are leveled by using it: E0, E2 and E3 appear to have similar mean values and variance, whereas E1 (p=0.6) and E4 (p=0.15) show visible differences to the control at 50 J/ml. Whereas E1 is not explained by the model and have to be discussed, E4 matches the forecast and is underpinned by the increase in bacterial cell release. That has to be carefully discussed. "There was a trend for increased POC release with increasing enzyme addition, and this trend was only broken by the control treatment (E0, given no enzymes)."

**Lines 284-288:** *There has been no physical transfer of organic matter between these analytical pools. A reduced aggregate stability may have for example, or increased release of biofilm fragments retained on the 1.5 µ glass filter, but this is a matter for discussion. It might be more useful to say here that it is reassuring that the SOC remaining in the sediment reflects what would be expected given the quantities extracted at 50*

*J... but of course it would (because you present relative fractions in preference to absolute concentrations). I am struggling to find a reason to retain this section. I think it better to delete lines.*
Our intention was to express that nearly the whole net POC differences E1-E0 and E4-E0 are related to variations in the HF, but not in fLF, oLF(100) and oLF(150). That will be included in lines 274-297.

**Line 289:** and Figure 2 These results have already been presented, it is not clear exactly what compounded estimate of error is being given, and besides, data were already presented in figure 1. Remove Figure 2. **Line 290:** This has already been presented, that one can add the non-significant results to the significant, and finds the same thing is nothing surprising or worthy of comment. Delete. **Line 291:** Clumsy sentence and repetition: delete first sentence. **And lines 291-293:** Released POM data may be evidence of this, and may not be - this is a matter for the discussion. Delete these lines.
Lines 289-293 will be deleted. Fig. 2 will be changed to mg POC /g dry soil and shortly described.

**Lines 293-296:** Delete section starting "The lower aggregate stability is indicated by a steeper gradient and on average in an...". Replace with "The addition of the highest enzyme concentration (E4) caused the release of about 40% more POM by mild sonication (50J ml-1) than occurred with the addition of the lowest concentration (E1). This was statistically significant at (p <0.05)." end of section.
Thank you very much. "At 50 J ml$^{-1}$ ultrasonic treatment results in an additional POC release of about 10% more POC compared to the control, whereas POC release is reduced by -18% in E1. The addition of the highest enzyme concentration (E4) caused the release of about 1/3 more POM by mild sonication (50 J ml$^{-1}$) than occurred with the addition of the lowest concentration (E1) (p=0.003).

**Line 302:** *In contrast here I think the relative increase in DNA release is a little understated. Yes it is useful to also give it as a percentage of total DNA extracted from the soil as you have done (Figure 3 - now rename to Figure 2), but perhaps in line 302 replace text "it is increased by about 3.5% to a value of 5.5% in the E4 scenario in comparison to the control" with 'While there was no difference in DNA concentrations suspended in the wash of control and low enzyme additions, treatment E4 caused an increase to more than double the DNA content of either E0 or E1."*
Thank you. Replaced by: "While there was no difference in relative DNA release in the wash of control and low enzyme additions, treatment E4 caused an increase to more than double the DNA content of either E0 or E1, which amounts to 5.6% of total DNA".

**Discussion**

**Lines 324-335:** First paragraph disorganised: it is an unpleasant jump to the model in the first sentence. Build up to it. It would be smoother if begin with the main result result, followed by your description of enzyme transport into the unsaturated pore space and discussion of others work E.g. "We found that increasing the quantity of enzymes applied to aggregates led to increased release of POM when aggregates were sonicated. Then describe the pore system (currently lines 325 326), then give your model of explanation "we present a model to explain the observed findings ..."
Thank you. First paragraph was replaced by:
"We found that increasing the quantity of enzymes applied to aggregates led to increased release of POC when aggregates were sonicated. This detachment is explained by the transport of α-glucosidase, β-galactosidase, DNAse and lipase into the unsaturated pore space. Consequently enzymes diffuse into the biofilm matrix, where structural components like polysaccharides, eDNA and lipids are digested as approved for diverse enzymes and enzyme targets in ecological and medical studies (Böckelmann et al., 2003; Walker et al., 2007). We utilize a simple spacial model to explain the observed findings: The biofilm bridges gaps between primary particles, connects them and builds a restructured pore system inside the aggregate (Fig. 4). As macromolecular biofilm components yield EPS as a viscoelastic structure (Sutherland, 2001), their digestion causes a loss in EPS viscosity and thereby should reduce aggregate stability. The effect is expected to grow with

increasing enzyme activity until the whole EPS matrix is dispersed."

**Lines 336-337:** *Delete the discussion of what is not being discussed.*
Done.

**Line 345:** 'de facto' is way too strong and encourages the reader think of examples to disprove this over-confident statement. E.g. it could have been caused by cell lysis. Delete 'de facto'.
Done.

**Line 352:** This is not the only possible explanation and further discussion with relevant literature is required. Might some of the C released from occluded POM and/or biofilm not have been detected in the filtered light fraction? – e.g. may have been present as smaller particulates or DOC? Also, DNA/cells/POM may not have been released without sonication. Include this. Current literature has more to offer. Add "Furthermore, we pre-incubated soils given 0.2 mM $NH_4NO_3$, and added further $NH_4NO_3$ with the enzyme application. Redmile-Gordon et al (2015) proposed that low C/N ratios of substrates available to soil microorganisms reduces cell specific EPS production rates, and may trigger microbial consumption of EPS to acquire C for cell-growth. The observations leading to this proposed dynamic were also found by addition of $NH_4NO_3$. In the present study, $NH_4NO_3$ was applied with all treatments including the control (which also received no C from enzyme provision). The resulting lowest C/N ratio in the control soils may itself have decreased the EPS, contributing to the higher than expected release of POM from the control soil with sonication at 50 J mL$^{-1}$, and the break in the trend for increasing POM release with increasing enzyme addition.

We now write: "Decreased POC release in E1 could be explained by pre-incubation of soil aggregates given 0.2 mM $NH_4NO_3$ and further addition of $NH_4NO_3$ with enzyme application. Redmile-Gordon et al. (2015) proposed that low C/N ratios of substrates available to soil microorganisms reduces cell specific EPS production rates, and may trigger microbial consumption of EPS to acquire C for cell-growth. The observations leading to this proposed dynamic were also found by addition of $NH_4NO_3$. In the present study, $NH_4NO_3$ was applied with all treatments including the control (which also received no C from enzyme provision). The resulting lowest C/N ratio in the control soils may itself have decreased the EPS, contributing to the higher than expected release of POM from the control soil with sonication at 50 J mL$^{-1}$, and the break in the trend for increasing POM release with increasing enzyme addition."

Further "Probably high enzyme concentrations dissolve biofilm structures that remain part of the coarse POM at low enzyme treatment, which results in underestimation of E4 POC release." was added in this paragraph.

**Lines 350-352, 390:** *Discussion about link between biofilm digestion and aggregate stability.*
Sentence in lines 350-352 "The incomplete biofilm digestion suggests, that the influence of biofilms on aggregate stability is larger than demonstrated in scenario E4." shifted to a later part of discussion. Previous reference to aggregate stability is replaced by biofilm digestion/POC release context, except in the spacial model.

**Line 353:** *Replace "The incomplete … ambiguously" sentence with "Nonetheless, biofilm detachment caused by E4 is still likely to be incomplete." And continue with "Slow enzyme diffusion..."*
Line 348-350 is replaced by "Hence, biofilm detachment caused by E4 is still likely to be incomplete." Continued with "Slow enzyme diffusion...".

**Lines 352, 356-367:** *This paragraph contains some useful information that should be retained for comparison of enzyme quantities added. However, the explanation drawing on enzyme activities in natural soils is not clear and needs re-thinking and re-writing. Actually, it seems the argument is flawed. You only observed effects when you increased enzyme activities well above 'natural' levels so on the contrary seems to support the hypothesis that diffusion factors ARE limiting (e.g. sorption to active surfaces). Suggest you cite the excellent review by (Burns et al., 2013) (see section 3.3; page 220).*
"Based on our calculations enzyme concentrations of mix E1 should be sufficient for total biofilm digestion within time of application (1h) – as far as there are no other factors reducing enzyme efficiency. As surveys of natural soils show enzyme concentrations up to mix E3 [Cooper and Morgan, 1981; Eivazi and Tabatabai, 1988; Acosta-Martinez and

Tabatabai, 2000], such factors might be reasonably assumed. This is underpinned by our results, that show the only increase in POC release in scenario E4 attended by only an incomplete cell release. After addition to the soil sample, enzymes must enter the EPS matrix by diffusion. Therefore it is assumed that parts of the enzymes probably do not reach the biofilm due to inhibited diffusion. Beside diffusion, sorption and decomposition could play a major role in reducing enzyme efficiency. Whereas turn-over rates of soil enzymes are not yet assessed, extended stabilization of active enzymes over time on soil mineral and organic surfaces is reported (Burns et al., 2013). This mechanism could explain immobilization of enzymes off the biofilm and high measured soil enzyme concentrations from literature in face of still existing biofilms. Due to this boundary conditions, quantification of the relation of enzyme concentration and POC release was not possible in this work, although there is a tendency for enhanced POC release." This information will be included in lines 356-367.

**Lines 368-370:** *It does not reinforce this, and if it does it conflicts with your model. If your model is correct it would only be found'after disruption of aggregates to release the oLF (as you observed at 50 J ml$^{-1}$; congruent with your model). It could also have been lost as soluble C, as mentioned above in reference to line 352 above. Delete 368 – 370.*
Correct. I referred to a POM occlusion only mediated by EPS, but in the model it seems very implausible to assume occluded POM, that is not bound by physico-chemical interactions. Deleted.

**Lines 378-383:** *Not statistically significant therefore remove this speculation. Statistically it is built on observations that can be reasonably expected by chance.*
Done.

**Line 384** *replace 'cumulation of LF carbon release overall energy level clarifies the alteration of soil aggregate stability' with 'The trend for increased of LF carbon release over increasing enzyme additions demonstrates an alteration of soil aggregate stability'.*
Thank you. Done.

**Line 385** – *results repetition. And* **lines 386-389** *Careful, you are discussing SOC (POM) release and aggregate stability as if you measured both independently, and focus drifts. I recommend you instead discuss the connection you propose (POM release being due to digestion of EPS which seems to prevent POM release by sonication alone up to 150 J ml-1 – and after more effectively separated from soil minerals by 50J sonication).*
See "New line of argument" at the end of this document.

**Line 395:** *Good point re enzyme metabolism, although 1 hour is not a lot of time for it, it would be useful to include a reference for rapid metabolism of enzymes/proteins. Add that the large additions of enzyme-C could be used as a C-source for microbial growth which is known to stabilise soil aggregates, e.g. (Watts et al., 2005). This is why total enzyme-C added should be included in your manuscript (suggest this is added to Table 3).*
"The applied enzymes have no relevant mass input to extractable POM. Even in case of complete adsorption to POM in only one fraction, highest enzyme concentration (E4) would result in additional 13.5 µg enzyme /g dry soil being <0.4% of the smallest extracted POM fraction.
Although enzyme concentration has no influence on extracted POC, addition of enzyme-C could be used as microbial metabolic C-source which is known to lead to soil aggregate stabilization (Watts et al., 2005; Tang et al., 2011). Soil turn-over rates of enzymes are not assessed (Burns et al., 2013). Fast metabolization of enzymes within 1 hour would hinder quantification of the relation of biofilm digestion and POC release by influencing aggregate stability during the experiment." This content will be connected to point (**Line 352**).

**Lines 407, 408:** better if you delete 'a 9000 fold of the E1 enzyme activity calculated from actual soil biomass to remove approximately // suggest replace '5.5% of the biofilm and no increase in FLF release, the pooled influence of the disregarded boundary conditions on enzymatic detachment efficiency is large' with

'5.5% biofilm removal indicated by DNA measurements coupled with no increase in fLF release, may suggest that the pooled influence of the disregarded boundary conditions on enzymatic detachment efficiency is large'.

As the role of fLF C is discussed regarding lines 368-370, the paragraph is replaced with: "Most of these restrictions are owed to the high complexity of the soil ecosystem. Enzymes were applied in concentrations four orders of magnitude higher than calculated from actual $C_{mic}$ and even 1-2 orders of magnitude higher than values from literature. Considering maximum 5.5% biofilm removal indicated by DNA measurements may suggest that the pooled influence of the disregarded boundary conditions on enzymatic detachment efficiency is large." Orders of magnitude are still noted to illustrate the probable range of influence of the disregarded boundary conditions.

**Lines 410-413:** *delete 'nonetheless' // replace 'Loss of aggregate stability' with 'Release of entrapped POM' // replace 'stabilisation' with 'stabilising' // Citation needed: suggest after 'stabilising agent of soil aggregates' to insert 'as discussed in a comprehensive review by Or et al. (2007)'. // Subsequent sentence, why limit to just natural ones? I suggest you replace 'Aggregate stability is influenced by the digestion of EPS components. Adapting this relation to natural soil ecosystems,"' with 'The apparent loss of aggregate stability caused by the digestion of EPS components in the present study suggests biofilm relevance in soil ecosystems.' And finish the discussion there.*

Paragraph replaced with "These results give insight in fundamental processes underlying aggregate stability. Release of occluded POM coupled with increased bacterial DNA release after treatment with high enzyme concentrations underpin the assumption that biofilm is a stabilising agent of soil aggregates as discussed in a review by Or et al. (2007). The apparent loss of aggregate stability caused by the digestion of EPS components in the present study suggests biofilm relevance in soil ecosystems e.g. in terms of soil-aggregate related functions like soil water dynamics, mechanical stability as well as rootability."

**Conclusion**

**Lines 414-417, 419-420, 422-423, 425, 425-427, 427, 431:** *Move this final part to the start of conclusions: "Our results suggest a change of biofilm composition due to a shift ..." // Already discussed, is weak, better to delete. // delete "and thereby enhances aggregate stability". Already discussed and now superseded by your two important sentences above this (first one suggested to be taken from discussion, lines 414 – 417). // Delete 'fLF' (these abstract technical distinctions are not appropriate for this statement). Continue with the condition i.e. "not to an increase in fLF release without physical disruption of aggregates by sonication." // replace SOC with POM (should already be defined) 427 delete the sentence starting "The bacterial DNA..." as discussed already; this does not withstand logical critique. // 'microbial communities' already are for various reasons, I think you mean the biofilm or EPS, EPS being relevant even when no biofilm can be observed … suggest you replace 'communities' with 'EPS dynamics'.*

New conclusion: "It was shown that EPS is a factor of aggregate stability. Our experimental results suggest that extracellular polymeric substance (EPS) contributes to occlusion and attachment of particulate organic matter (POM) in soil aggregates. The application of a highly concentrated mix of α-glucosidase, β-galactosidase, DNAse and lipase is related to a detachment of POM from a stable to a more fragile binding structure, but not to an increase in POM release without physical disruption of aggregates by sonication. The pattern of measured POC release and additional bacterial DNA release points to an intra-aggregate fixation of POM by enzyme targets. A loss of EPS integrity could therefore cause a detachment of soil organic matter, not only in the laboratory but also in natural soil ecosystems. Our results further suggest that a change of biofilm composition probably due to a shift in microbial population structure may alter soil aggregate stability. On macro-scale this could affect soil compactibility, erodibility, water transport, retention and aeration regime, rooting depth and the occlusion of soil organic carbon. This, in conclusion, invites to behold soil EPS dynamics as a factor of sustainable land use."

**Figures and Tables**

**Figure 4:** edit caption – you are not showing 'biofilm structure' – this is 'aggregate structure' replace accordingly.
Caption changed to "Proposed model of aggregate structure: ..."

**Table 3:** Add quantity of enzyme-C added to enable judgement of substrate utilisation by soil microbial biomass.
Quantities added.

**Table 3:** column E0: should the q value not be zero? Otherwise why are the enzyme activities different from column E1?
Yes. Thanks.

**Furthermore ...**
… there are also some points I have to answer back.

**Line 38:** *insert 'and' before 'is an integral'*
That doesn't fit in this place.

**Line 56:** *delete '.'*
There is an end of sentence and the references are related to the whole paragraph.

**Line 173:** *use large 'C' for carbon*
c in $c_{cell}$ means "concentration"

**Line 409:** *Insert sentence: 'Conversely, or in addition to the above, complete biofilm removal may have been achieved, however as the model (figure 4 – now figure 3) proposes, POM would not be released until the retaining aggregates were disrupted by disruptive physical forces such as those caused by sonication.' (Kaiser and Berhe, 2014)*
As only 5.5% of the bacterial DNA are removed after enzymatic treatment, it seems implausible to expect complete biofilm detachment. Further, point (**Lines 368-370**).

**New line of argument**
Line of argument will be restructured in the following way (e.g. to avoid repetitions): Discussion of POM release (increase in E4, decrease in E1, tendency, p-values but no significance level) – discussion of bacterial DNA release – discussion of of the relation of both (EPS as enzyme target) – discussion of the explanatory power of (small) POM release and of its usability for aggregate stability measurement in similar soil samples – "A more quantitative analysis of the relation of enzymatic EPS detachment and POM release would require more replicate samples and probably inclusion of soils from different land use. However, this was beyond the scope of the present study."

*Best regards,*
*Frederick Büks*

**Enzymatic biofilm detachment causes a loss of aggregate stability in a sandy soil.**

**F. Büks[1] and M. Kaupenjohann[1]**

[1] Chair of Soil Science, Department of Ecology, Technische Universität Berlin.

*Correspondence to:* F. Büks (frederick.bueks@tu-berlin.de)

**Final response to Referee2**

*Dear Referee2.*
*Thank you very much for reviewing. In the following I will try to answer your important comments and to clear the objections.*

**1.a.** *Section 2.2: confusingly written maths section*
Pooling of equations to a single one is a space saving way to show these manifold steps of converting concentrations of biofilm components to the final value of needed enzyme units. I will place each single step in the supplements. The paragraph beginning in line 163 will be revised to clarify the following: Literature show a wide range of enzyme-target concentrations in different soils. As we do not know target concentrations of our soil (due to a lack of extraction methods), we considered the largest published concentrations to find existing effects. Further as target molar masses vary as well, here we choose the smallest mass. Both conduce to a "worst-case" point of view with maximum enzyme targets.

**1.b.** *Section 2.2: poor justification of numbers used: Eg the supposed soil bulk density number seems odd, as this can be measured for field core samples and be recreated to field soil density. Otherwise explain the assumption for this particular experiment as normal dried and sieved soil without repacking does not get to this density.*
Different samplings during the field experiment showed soil bulk densities of 1.4 g/cm$^3$. These values are normal for a sandy silt (Su3) [Chaudhari et al., 2013], that is used in this experiment.
For scenario E1 soil bulk density is irrelevant because $c_{cell}$ and therefore target maxima were estimated from $c_{mic}$. For scenario E2 and the following we measured a minor soil bulk density in a sample of soil aggregates (~1.15 g/cm$^3$). On the other hand, biofilm populations are mentioned to be mainly located in soil aggregates [Nunan et al., 2003]. Therefore – following our "worst-case"-approach – we used the bulk density of the original soil to estimate maximum target values.

**1.c.** *Section 2.2/2.3: poor justification of numbers used: The 'scenarios' have been explained (though could be improved in clarity) but do not actually contain any information regarding the technical set up.*
*How much enzyme activity units were applied?*
Enzyme units are listed in table 3.
*What was the level of purity of the enzyme preparations?*
Enzyme purity is guaranteed by the producer (data sheets of product numbers Sigma-Aldrich: G0660, G5635, L0382 and D5025).
*How where the enzymes added? Was there mixing involved?*
Enzymes were added as described in section 2.4. Enzyme solutions were vortexed and

than added to aggregate samples as described in section 2.4.

*There is a severe lack of information, especially as the whole manuscript depends on contact of these enzymes with EPS materials. How have the authors assured that these enzymes have reached the materials processed further?*

Contact of enzymes and EPS on the micro-scale were not demonstrated directly. Contact of enzymes to EPS can be assumed as the whole enzyme solution was absorbed by the soil aggregates. Fine pores (already filled with ARW from pre-incubation) in contrast need to be supplied by diffusion, that is probably inhibited. However, enzymes are able to diffuse into the EPS within 1 hour, as described by Böckelmann et al. (2003). Thus, observed effects are not quantitative, but qualitative. We tend to express the more cautious position "Enzymatic treatment causes an increased release of POM after sonication" to include uncertainties about enzyme contact to targets.

**1.d.** *Section 2.2/2.3: the E4 scenario seems to suggest a large excess of enzymes was applied. How have the authors ensured that such a large excess is not damaging to resident live microbial cells? E.g. a large excess of lipase may affect the membrane integrity of cells. This may in turn impact on DNA quantification without actually directly affecting soil aggregate stability.*

Cell membranes are built of phospholipids. We used purified lipase from porcine pancreas. Lipases are cutting fatty acids off e.g. glycerol, but are unable to cut fatty acids from phospholipids (as phospholipases do).

**1.e.** *Section 2.3: information/studies on basal respiration at 30C/37C, the temperature of the actual experiments performed, are missing.*

Respiration data are collected at 20°C in another experiment using the same soil, where basal respiration was reached after 2 days. Therefore we concluded 3 days as sufficient to reach basal respiration at even higher temperatures.

**1.f.** *Section 2.4: this experiment was performed on a separate soil incubation experiment within kit tubes. The experiment should however have been performed on subsamples taken from the experiment in 2.2/2.3 as the conditions in (closed?) kit tubes are very different from regular soil incubations. The authors attempt to link the results from both experiments, which in my opinion is not warranted as the experiments have been performed under different conditions.*

A direct subsampling from the aggregate stability experiment to perform the DNA experiment was rejected due to its destructive capability regarding aggregates. In turn, temperature, substrate, pH and water content of the tube experiment were similar to the incubation of samples for the measurement of aggregate stability. Further differences were disregarded. As part of our hypothesis, the link between both increase in POM release and bacterial cell release can explained causally.

**1.g.** *Section 2.4: for especially scenario E4, with an apparent excess of enzymes including DNase, I am surprised to see the authors report successful DNA purification. How have the authors achieved DNA purification in the presence of excess DNase? Idem for the scenarios with lower amount(s) of DNAse added?*

During incubation DNAse only digest free DNA but not DNA within bacterial cells of the biofilm. Later the pooled wash solution contains most of the DNAse. After centrifugation, this solution was discarded, whereas the bacterial pallet was resuspended in 200 µl ARW. At this stage, bacterial cells are still intact and immune to DNAse, whereas DNAse is diluted and hindered by high buffer ion concentrations. After mechanical cell lysis PPS (Protein Precipitation Solution) was added leading to e.g. precipitate DNAse. All steps of DNA extraction were conducted on ice to strongly reduce enzyme activity.

**2.a.** *The results of soil stability/SOM measurements indicate that none of the 'scenarios' are significantly different from the control experiment. The only significant difference the authors report concerns between treatment results, which leaves me wondering about the relevance of the whole study.*

Even if there is no significant difference in aggregate stability, unfulfilled expectations (as e.g. a dramatic loss in aggregate stability after enzymatic treatment) do not minder the

relevance of a study. In addition – without any attempt to prettify our results – p-values <0.05 as the limit for significance is a convention. From my point of view there is a tendency of increasing POM release (p=0.1, 5 parallels) in E4, and a tendency to decrease (p=0.06, 5 parallels) in E1 compared to the control. The first one fits to our model, the second one does not. Both tendencies are visible and have to be explained under the restriction of being small.

**2.b.** *The results shown in Figure 2 have been reported without statistical analyses on significant difference. Please include statistical analyses on significant difference between control and treatments. The figure's error bars of the control and the experimental treatments could suggest that differences between control and treatment scenarios are unlikely to be significant, leaving doubt about the experiment's relevance and study design.*
Good idea. I will do this. Thereby, y-axis of figure 2 will be converted to mg POC /g dry soil.

**2.c.** *Figure 3 is missing a control on DNA present in the added enzyme mixtures. Can the authors ensure that the DNA extracted and amplified is not derived from the enzyme preparations added? Especially scenario E4 might lead to addition of a lot of DNA.*
We do not have this data. ARW for stock solutions and dilutions include ultrapure water and were autoclaved. Remaining free DNA strand amount is assumed to be far below soil DNA concentration and most probably digested by DNAse in stock and dilutet solutions. Further possible DNA additions were similar between variants and related to blind values.

**2.d.** *Figure 3: In contrast to the above, DNase is added in the scenarios, which should then lead to degradation of DNA present in the samples. Can the authors therefore please clarify the puzzling details of this experiment?*
Until mechanical cell lysis, extracellular DNA including eDNA from EPS is digested by DNAse. As mentioned in 2.c. small amounts of additional DNA are supposed to be irrelevant and the bulk of free DNA is rejected by washing. E4 shows the highest DNA release, although undigested biofilm in low enzyme treatments could increase DNA-concentration via centrifugation to the pallet. That could probably point to an underestimated additional DNA release in E4. It underlines the only qualitative approach of this experiment.

**2.e.** *Figure 3: Can the authors please provide (control) data on (expected) cell lysis from treatments, esp E4? This will enable untangling of results due to lysis and any EPS - biofilm effect on soil aggregation.*
DOC release from bacterial cells due to enzymatic treatment and ultrasonication was not quantified. This DOC is most probably removed by repeated washing during density fractioning. Measuring the distribution of remaining bacterial DOC among fLF, oLF and HF is impossible by method. However, visibly increasing POM release in E4 (see Line 30, Final response to Marc Redmile-Gordon) points to a negligible effect of bacterial DOC sorption on measured C.

**3.a.** *The significant in – between - treatment results are given too much focus and attention, especially in the knowledge that none of the treatments were significantly different to controls. The majority of the conclusions drawn are not supported by the actual data provided.*
**and 3.b.** *Line 390 '... our results give a qualitative evidence for the influence of biofilms on aggregate stability...' This conclusion is not supported by the data provided.*
See point 2.a. We propose careful line of argument including a statement of insignificance and a discussion of tendencies in face of p-values nearby p=0.05.

**3.b.** *Figure 4: this diagram can be omitted.*
As figure 4 illustrates the model, we prefer to retain it.

*Best regards,*
*Frederick Büks*

---

## Referee Report (RR1)

**Marc Redmile-Gordon (PhD)**
**Dept. of Sustainable Soils and Grassland Systems**
**Rothamsted Research**
**Harpenden**
**Hertfordshire**
**AL5 2JQ**

**Enzymatic biofilm digestion in soil aggregates facilitates the release of particulate organic matter by sonication**

**Dear Editor and Authors,**

**I am writing regarding the second review of Manuscript "soil-2015-87-manuscript-version3.pdf"**

**In my opinion the paper has been much improved, and the responses to my comments are well reasoned. I think this is an interesting development in an exciting area of soil science. However, before I would recommend publication, I do think there are some important changes that a) have not been done as it is claimed (presumable just file version errors), but b) do need to be done.**

**This is to refine the clarity of the message in the Author's well-reasoned 'new line of argument' and to clarify how NH4NO3 may have explained the deviation from the expected results relative to the control (which taken collectively would mean results do not contradict the aggregate stabilisation mechanism proposed by the authors).**

**Attending to some specific responses:**

Authors [8] We now write: "Decreased POC release in E1 could be explained by pre-incubation of soil aggregates given 0.2 mM $NH_4NO_3$ and further addition of $NH_4NO_3$ with enzyme application. Redmile-Gordon et al. (2015) proposed that low C/N ratios of substrates available to soil microorganisms reduces cell specific EPS production rates, and may trigger microbial consumption of EPS to acquire C for cell-growth. The observations leading to this proposed dynamic were also found by addition of $NH_4NO_3$. In the present study, $NH_4NO_3$ was applied with all treatments including the control (which also received no C from enzyme provision). The resulting lowest C/N ratio in the control soils may itself have decreased the EPS, contributing to the higher than expected release of POM from the control soil with sonication at 50 J $mL^{-1}$, and the break in the trend for increasing POM release with increasing enzyme addition."

**MRG: The above is fine, but, the actual messages in manuscript version 3 are fragmented.**

**On the contrary to what is suggested above, the paragraph above is not what appears in the latest version of the manuscript [lines 421 to 435]. This might be ok if the section clearly retained the same meaning. But the narrative is not clear. I suggest the authors present the (uninterrupted) paragraph that they state has been written above. The manuscript would read more easily if it was done as claimed above, and with the interfering sentences about enzymes being used as a carbon source grouped together, removed, and placed in a subsequent paragraph. Thus dealing with the impact of N first.**

**Accordingly, I suggest:**

**1) delete distracting line over 418-419: "Whereas E4 matches the forecast of releasing more POM than the control scenario" (this is a repeated message – It is already clear that E4 is in line with the model).**

**2) to move the distracting but important statement currently at line 429-431 to a subsequent paragraph (see later suggestion)** "Enzyme C in E1 to E4 could be used as microbial C source. The addition of SOC is known to lead to soil aggregate stabilization (Watts et al., 2005; Tang et al., 2011) and withdraw the effect of reduced C/N ratio."

**3) Please also correct the above sentence as is not strictly correct, - i.e. the cited authors do not' Add SOC', and more importantly this changes the C/N ratio, (not removing its significance). The Authors do not state the resulting C/N for E1, and this is fine, but it cannot be presumed that it 'withdraws the effect of C/N ratio'. It changes the C/N ratio of labile substrate… but to what? For simplicity in addition to moving it I suggest it is just changed to:**

*"Enzyme C in E1 to E4 could be used as microbial C source. The addition of C increases the C/N ratio and has been shown to lead to soil aggregate stabilization (Watts et al., 2005; Tang et al., 2011)."*

**And then**

**4) delete 'in contrast, the retention of' at line 431: As now there will be no need for 'in contrast' as the narrative is not confounded by the additional statement found in the present version of the manuscript [v3] at lines 429-431. 'The retention of' is also confusing, I think I know what is in mind, but it is still confusing and makes the discussion unclear.**

**i.e. [lines 431 435]**  the lowest C/N ratio in the control soils may itself have sustained EPS consumption and repressed reconstruction of the EPS, contributing to the higher than expected release of POM from the control soil with sonication at 50 J mL$_{-1}$ and the break in the trend for increasing POM release with increasing enzyme addition.

**Also) Paragraph break needed around line 435. I think 1) Delete 'However'. 2) continue with 'Decay rates of enzymes… …carbon source' 3) insert** *"Enzyme C in E1 to E4 could be used as microbial C source. The addition of C increases the C/N ratio and has been shown*

*to lead to soil aggregate stabilization (Watts et al., 2005; Tang et al., 2011)."* **Then perhaps 4) continue with "Under certain conditions POM carbon…**

**I hope you find these suggestions helpful.**

**Next follows another example where the manuscript has not been changed as described (which makes things a little difficult for me).**

[9] Further "Probably high enzyme concentrations dissolve biofilm structures that remain part of the coarse POM at low enzyme treatment, which results in underestimation of E4 POC release." was added in this paragraph.

**Searches for components of the sentence in the script yields no results – I also do not know what this sentence means and I do suggest it is not re-added…**

**Please make minimal changes and document them fully.**

Authors [12] "The applied enzymes have no relevant mass input to extractable POM"

**Agreed.**

**Minor corrections:**

*line 67 and 73 Typo's ? (consider removing full stops?)*
*217 – 'silT'? (though I do like 'silky' as a description)*
*Results: Paragraph formatting (spaces) needs attention.*

**It has been a pleasure to learn of your work.**

**Best wishes and regards,**
**Marc**

---

## Author Response (AR2)

**Enzymatic biofilm digestion in soil aggregates facilitates the release of particulate organic matter by sonication**

**F. Büks[1] and M. Kaupenjohann[1]**

[1] Chair of Soil Science, Department of Ecology, Technische Universität Berlin.

*Correspondence to:* F. Büks (frederick.bueks@tu-berlin.de)

*Dear Mr Redmile-Gordon.*

*Thank you again for your helpful suggestions. Please pardon the puzzling corrections of my last revision and let me know if I missed a point.*
*Best regards,*

*Frederick Büks*

**Contents**
- done revisions
- revised manuscript (corrections are marked in yellow)
- former manuscript (corrected passages are marked in yellow)

**1)** "Whereas E4 matches the forecast of releasing more POM than the control, scenario E1 shows a reduced release by -2.8% and the DNA release remains unchanged compared to the control. This decrease in the 50 J ml$^{-1}$ fraction is related to an increase in the sediment fraction and cannot be explained by the model (Fig. 1)." deleted. Next sentence: " Probably this could be explained ..."

**2 and 3)** Lines 429-31 were changed to "Enzyme C in E1 to E4 could be used as microbial C source. The addition of C increases the C/N ratio and has been shown to lead to soil aggregate stabilization (Watts et al., 2005; Tang et al., 2011)." and moved behind "... increasing enzyme addition" (Line 435).

**4)** "in contrast, the retention of ..." and "However ..." were deleted.

[revised manuscript text omitted]